# OSCAR: One-Step Diffusion Codec Across Multiple Bit-rates

**Jinpei Guo**[1,2], **Yifei Ji**[2], **Zheng Chen**[2], **Kai Liu**[2],
**Min Liu**[1], **Wang Rao**[1], **Wenbo Li**[3], **Yong Guo**[4], **Yulun Zhang**[2*]
[1]Carnegie Mellon University, [2]Shanghai Jiao Tong University,
[3]Joy Future Academy, [4]South China University of Technology

## Abstract

Pretrained latent diffusion models have shown strong potential for lossy image compression, owing to their powerful generative priors. Most existing diffusion-based methods reconstruct images by iteratively denoising from random noise, guided by compressed latent representations. While these approaches have achieved high reconstruction quality, their multi-step sampling process incurs substantial computational overhead. Moreover, they typically require training separate models for different compression bit-rates, leading to significant training and storage costs. To address these challenges, we propose a **o**ne-**s**tep diffusion **c**odec **a**cross multiple bit-**r**ates. termed **OSCAR**. Specifically, our method views compressed latents as noisy variants of the original latents, where the level of distortion depends on the bit-rate. This perspective allows them to be modeled as intermediate states along a diffusion trajectory. By establishing a mapping from the compression bit-rate to a pseudo diffusion timestep, we condition a single generative model to support reconstructions at multiple bit-rates. Meanwhile, we argue that the compressed latents retain rich structural information, thereby making one-step denoising feasible. Thus, OSCAR replaces iterative sampling with a single denoising pass, significantly improving inference efficiency. Extensive experiments demonstrate that OSCAR achieves superior performance in both quantitative and visual quality metrics. The code and models are available at https://github.com/jp-guo/OSCAR.

## 1 Introduction

With the explosive growth of multimedia content, the need for efficient transmission and storage of high-resolution images has become increasingly critical. Traditional image compression has long relied on manually crafted codecs [53, 7, 11], which tend to falter at very low bit rates ($\leq 0.2$ bpp), where blocking and ringing artifacts severely degrade perceptual quality. Recent research has shifted towards learning-based transform-coding systems [5, 14, 22] that jointly optimize an end-to-end rate–distortion (R–D) objective [44]. Although these neural codecs generally outperform the handcrafted compression algorithms, the focus on minimizing R–D inevitably sacrifices fine-grained realism at extreme compression bit-rates [9], producing images that appear synthetic or over-smoothed. In addition, supporting multiple compression bit-rates generally means training multiple models, each optimized with a rate–distortion loss weighted by a different Lagrange multiplier ($\lambda$). Maintaining one model per $\lambda$ dramatically inflates both training time and storage requirements.

Recent breakthroughs in diffusion models [45, 24, 46], most notably large-scale text-to-image latent diffusion models (LDMs) [42], have revealed powerful generative priors that can be repurposed for a wide range of downstream vision tasks [6, 10, 20]. Building on this insight, diffusion-based codecs [47, 32, 34] have been proposed for image compression. These models typically perform

---

*Corresponding author: Yulun Zhang, yulun100@gmail.com

39th Conference on Neural Information Processing Systems (NeurIPS 2025).

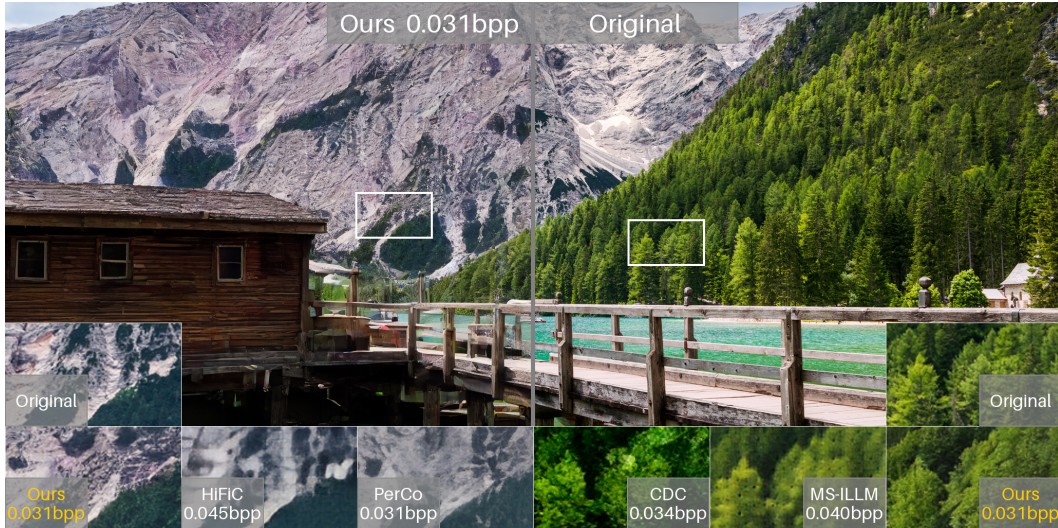

Figure 1: Qualitative comparison. OSCAR is capable of reconstructing complex textures with high realism. OSCAR effectively reconstructs complex textures with high realism.

quantization in the latent space to enable compression, achieving lower distortion at a given rate than conventional CNN- or Transformer-based methods. Some works [12] further integrate pretrained LDMs with a vector-quantization mechanism [51, 17], allowing the compression rate to be predetermined via a fixed hyper-encoder. Their practicality, however, is still hindered by two issues: (i) the multiple denoising iterations are required at inference time, which impose substantial inference computational overhead; and (ii) achieving multiple bit-rates still requires training separate diffusion networks. Since diffusion models typically have a larger model size than conventional learned codecs, this one-model-per-rate strategy further amplifies both training and storage costs. These constraints limit real-world deployment, particularly in resource-constrained scenarios.

To address the aforementioned challenges, we propose **OSCAR**, a **o**ne-**s**tep diffusion **c**odec **a**cross multiple bit-**r**ates. Unlike previous approaches [58, 34, 12] that start from Gaussian noise, we introduce the concept of *pseudo diffusion timestep*, bridging the quantization process and the forward diffusion. This idea is inspired by a classical signal processing result that quantization noise, especially at higher bit-rates, can be approximated as additive Gaussian noise [8]. Formally, we posit that the quantized latents $\tilde{\mathbf{z}}$ can be decomposed as $\tilde{\mathbf{z}} = \sqrt{\bar{\alpha}_t}\mathbf{z}_0 + \sqrt{1 - \bar{\alpha}_t}\boldsymbol{\epsilon}$, where $\boldsymbol{\epsilon}$ is approximately Gaussian noise and statistically independent of $\mathbf{z}_0$, $\bar{\alpha}_t$ is the diffusion noise schedule, and $t$ is the pseudo diffusion timestep that depends on the bit-rate. Under the assumption that $\boldsymbol{\epsilon}$ is orthogonal to $\mathbf{z}_0$ in expectation, the expected cosine similarity between $\tilde{\mathbf{z}}$ and $\mathbf{z}_0$ satisfies $\mathbb{E}[\text{sim}(\tilde{\mathbf{z}}, \mathbf{z}_0)] \approx \sqrt{\bar{\alpha}_t}$ (Please refer to Appendix for a full derivation). To make this relationship more consistent, we adopt a representation alignment training between $\tilde{\mathbf{z}}$ and $\mathbf{z}_0$. We empirically observe that this training paradigm enables their cosine similarity to converge to a fixed value for each bit-rate. Thus, by matching measured cosine similarities to the theoretical form, each bit-rate $r$ can be directly mapped to a pseudo diffusion timestep $t$. Our experiments provide strong support for this modeling, as the measured distribution of $\boldsymbol{\epsilon}$ closely follows the assumed Gaussian distribution.

We further illustrate this perspective in Fig. 2(a), where the quantized latent features can be interpreted as the intermediate diffusion states along diffusion trajectories. Remarkably, even under extreme compression, the cosine similarity between $\tilde{\mathbf{z}}$ and $\mathbf{z}_0$ stabilizes to a high value once representation alignment is applied. This suggests that the $\tilde{\mathbf{z}}$ retains rich structural information and is amenable to one-step denoising. Since each bit-rate corresponds to a distinct pseudo timestep, the reconstruction process can be unified: A single diffusion model can denoise quantized latents at different compression bit-rates in one pass, conditioned on their respective pseudo diffusion timesteps. As shown in Fig. 2(b), OSCAR achieves a strong quality–efficiency tradeoff. Furthermore, Fig. 1 shows that OSCAR can faithfully restore fine textures and perceptually coherent details at low bit-rates.

Our main contributions are summarized as follows:

- We propose a novel and efficient one-step diffusion codec OSCAR, which reconstructs compressed latents into high-quality images in a single denoising step. Unlike existing

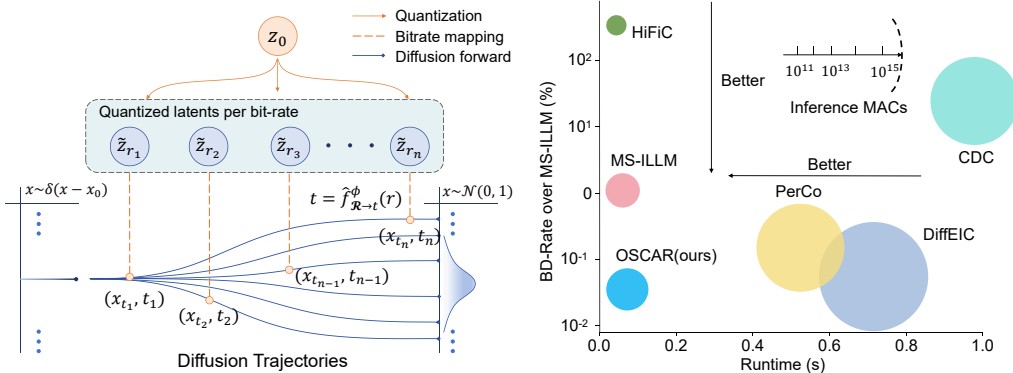

(a) Illustration of mapping between bit-rates and timesteps  (b) BD-Rate comparison on Kodak dataset.

Figure 2: Left: Our method bridges quantization and diffusion by mapping bit-rates to pseudo diffusion timesteps. Right: BD-rates measured by DISTS [16], with circle radius indicating multiply-accumulate operations (MACs). OSCAR achieves a favorable quality-efficiency tradeoff.

> diffusion codecs that require multiple iterative steps, our approach enables rapid and high-fidelity decoding. To the best of our knowledge, this is the first work to explore one-step diffusion for image compression.

- We propose a unified network design that supports compression at multiple bit-rates. By bridging quantization and forward diffusion, our approach enables a single generative model to handle diverse bit-rates without the need to train separate models. This unified design reduces storage requirements and simplifies training.

- Through extensive experiments, we demonstrate that OSCAR exhibits clear advantages over traditional codecs and learning-based methods in both quantitative metrics and visual quality. Moreover, it achieves competitive performance compared to multi-step diffusion codecs, while operating orders of magnitude faster.

## 2 Related Work

### 2.1 Neural Image Compression

Deep learning has enabled neural image compression methods to outperform traditional codecs such as JPEG [53], BPG [7], and VVC [11]. Pioneering work [4] introduced end-to-end autoencoders for learned compression, followed by efforts [5, 38, 14, 22] to jointly optimize rate and distortion through improved latent representations. Yet, distortion-based losses (e.g., MSE) often lead to distribution shifts between reconstructed and natural images [9], motivating research [3, 23, 1] to enhance perceptual quality. HiFiC [37] explores the integration of GANs [19] with compression. MS-ILLM [39] improves reconstruction quality through improved discriminator design. Other methods leverage textual semantics [30], causal modeling [21], or decoding-time guidance [1] to further boost perceptual fidelity. However, the aforementioned methods typically require training separate models for each bit-rate, which results in substantial overhead.

To address the inefficiency of training separate models for each bit-rate, recent research has explored adaptive compression strategies that enable flexible bit-rate control. In particular, progressive compression [50, 36, 59] has attracted significant interest for its ability to produce scalable bitstreams. DPICT [31] encodes latent features into trit-plane bitstreams, allowing fine-grained scalability. CTC [26] extends this idea by introducing context-based trit-plane coding. Meanwhile, variable-rate approaches [13, 54] have demonstrated effectiveness by dynamically adjusting scalar parameters or employing conditional convolutions to accommodate diverse quality levels. Despite their flexibility, these methods tend to suffer from notable quality degradation at very low bit-rates.

### 2.2 Diffusion Models

Diffusion models are powerful generative frameworks that transform random noise into coherent data through a multi-step denoising process [24]. Denoising Diffusion Implicit Models (DDIM) [46]

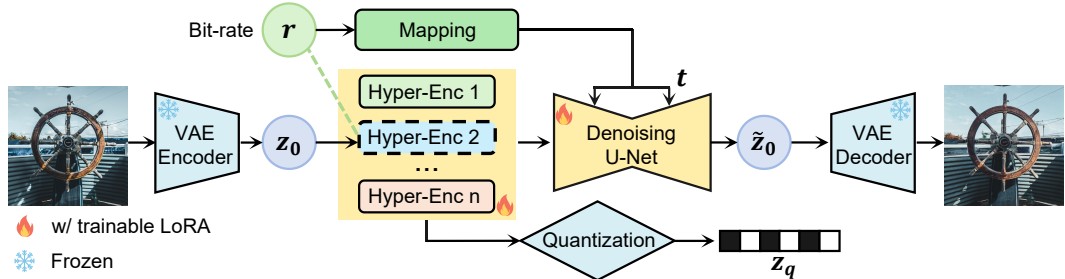

Figure 3: Overview of OSCAR. Given an input image, the VAE encoder first maps it to the latent space. The latent is then processed by a bit-rate-specific hyper-encoder and quantized via the corresponding codebook. During decompression, the bit-rate is mapped to a pseudo diffusion timestep, and the quantized features are denoised by a U-Net conditioned on this timestep to recover the latent. Finally, the VAE decoder reconstructs the image from the restored latent.

accelerate this process without compromising quality, influencing many subsequent advances. Latent Diffusion Models (LDMs) further improve efficiency by operating in a compressed latent space. Stable Diffusion (SD) [42] advances the LDM architecture into a scalable text-to-image system, enabling flexible multimodal conditioning while maintaining strong image fidelity. In our work, we build on SD to benefit from its strong image generation capabilities.

Beyond image synthesis, diffusion models have also been explored in image compression [57, 40]. Early work [24] explored diffusion-based compression through reverse channel coding, with an emphasis on rate-distortion trade-offs. Follow-up studies [47] extended this line by applying the technique to lossy transmission of Gaussian samples. Recent approaches further leverage large-scale pretrained diffusion models to enhance performance [52]. CDC [58] replaces the VAE decoder with a conditional diffusion model, reconstructing images via reverse diffusion conditioned on a learned content latent. PerCo [12] adopts a VQ-VAE-style codebook for discrete representation and incorporates global textual descriptions to guide the iterative decoding process. DiffEIC [34] uses a lightweight control module to guide a frozen diffusion model with compressed content. While these models achieve strong results, their multi-step denoising process remains computationally expensive. Improving sampling efficiency is therefore essential for practical diffusion-based image compression.

## 3 Method

### 3.1 Preliminaries: One-Step Latent Diffusion Model

Latent Diffusion Models (LDMs) [42] operate in a lower-dimensional latent space rather than pixel space, enabling more efficient training and generation. The forward diffusion process perturbs a clean latent variable $\mathbf{z}$ by injecting Gaussian noise. At timestep $t$, the corrupted latent is given by:

$$\mathbf{z}_t = \sqrt{\bar{\alpha}_t}\mathbf{z} + \sqrt{1 - \bar{\alpha}_t}\,\boldsymbol{\epsilon}, \quad \boldsymbol{\epsilon} \sim \mathcal{N}(\mathbf{0}, \mathbf{I}), \tag{1}$$

where the noise schedule is encoded as $\bar{\alpha}_t = \prod_{i=1}^{t}(1 - \beta_i)$, and $\beta_i$ is the variance. The reverse process aims to progressively recover the original latent through a learned denoising network [24, 46], typically implemented as a U-Net [43] structure in LDMs. Standard diffusion models perform multi-step denoising via a Markov chain, where each reverse step is modeled as:

$$p_\theta(\mathbf{z}_{t-1}|\mathbf{z}_t) = \mathcal{N}(\mathbf{z}_{t-1} \mid \boldsymbol{\mu}_\theta(\mathbf{z}_t, t), \beta_t\mathbf{I}), \tag{2}$$

with $\boldsymbol{\mu}_\theta$ derived from a neural noise estimator $\boldsymbol{\epsilon}_\theta(\mathbf{z}_t, t)$. In the one-step variant, a single denoising pass directly estimates the clean latent $\hat{\mathbf{z}}_0$ from a noisy input $\mathbf{z}_t$:

$$\hat{\mathbf{z}}_0 = \left(\mathbf{z}_t - \sqrt{1 - \bar{\alpha}_t}\,\boldsymbol{\epsilon}_\theta(\mathbf{z}_t, t)\right)/\sqrt{\bar{\alpha}_t}. \tag{3}$$

This formulation enables efficient reconstruction by collapsing the multi-step diffusion process into a single inference step. By viewing quantized latent features as intermediate diffusion states, our model leverages this formulation to efficiently recover clean latent representations.

### 3.2 OSCAR Overview

An overview of our OSCAR is shown in Fig. 3, and the corresponding pseudocode is provided in the Appendix. The training of OSCAR consists of two stages. In the first stage, we train multiple bit-rate-specific hyper-encoders to align the quantized latent representations $\tilde{\mathbf{z}}$ with the original latents

$\mathbf{z}_0$. We empirically find that the cosine similarity between $\tilde{\mathbf{z}}$ and $\mathbf{z}_0$ stabilizes after this stage, allowing us to construct a mapping function $\hat{f}_{\mathcal{R} \to t}^{\phi}(\cdot)$ that bridges bit-rates and pseudo diffusion timesteps, where $\mathcal{R}$ is the predefined bit-rate set. In the second stage, we jointly fine-tune the SD U-Net and all hyper-encoders to improve reconstruction fidelity. Since our compression and decompression are entirely performed in the latent space, we freeze the parameters of the SD VAE during training.

At inference time, the OSCAR pipeline can be divided into two phases: compression and decompression. During compression, given an input image $\mathbf{I}$ and the desired bit-rate $r \in \mathcal{R}$, we first obtain its latent representation $\mathbf{z}_0$ via the frozen VAE encoder $\mathcal{E}_\theta$. The latent is then quantized using a bit-rate-specific hyper-encoder $\mathcal{Q}_\phi$ to produce the quantized latent representation $\tilde{\mathbf{z}}$:

$$\mathbf{z}_0 = \mathcal{E}_\theta(\mathbf{I}), \quad \tilde{\mathbf{z}} = \mathcal{Q}_\phi(\mathbf{z}_0; r). \tag{4}$$

During decompression, we interpret $\tilde{\mathbf{z}}$ as the intermediate diffusion state. A pseudo diffusion timestep is assigned for each bit-rate, and the denoising network $\mathcal{DN}_\theta$ (i.e., the SD U-Net) restores fine-grained latent details to produce $\hat{\mathbf{z}}_0$. The reconstructed latent is subsequently decoded by the VAE decoder to obtain the final image. We denote the full reconstruction pipeline—including the hyper-encoder, denoising network, and decoder—as a generator $\mathcal{G}_{\phi,\theta}$. The reconstructed latent features and image can thus be expressed as:

$$\hat{\mathbf{z}}_0 = \mathcal{DN}_\theta(\tilde{\mathbf{z}}; \hat{f}_{\mathcal{R} \to t}^{\phi}(r)), \quad \hat{\mathbf{I}} = \mathcal{G}_{\phi,\theta}(\mathbf{z}_0). \tag{5}$$

### 3.3 Stage 1: Learning Quantization Hyper-Encoders

**Bit-rate-specific hyper-encoders.** While latent features benefit from low spatial resolution and are well-suited for compression, they still require significant storage, as they encapsulate high-dimensional semantic information [42, 41]. To address this, we apply hyper-encoders to quantize the latent representations. Given an input image, OSCAR first uses the Stable Diffusion (SD) VAE encoder to extract the latent representation $\mathbf{z}_0$. This step reduces spatial resolution (e.g., a $1024 \times 1024$ image yields a latent of shape $4 \times 128 \times 128$). To further compress $\mathbf{z}_0 \in \mathbb{R}^{4 \times h \times w}$, we employ a lightweight hyper-encoder that downsamples and quantizes it into $\mathbf{z}_q \in \mathbb{R}^{M \times h' \times w'}$, where $M$ is the dimension of the quantization space. Specifically for the architecture, the front-end of the hyper-encoder consists of several residual blocks, followed by a vector quantization (VQ) codebook. With down-

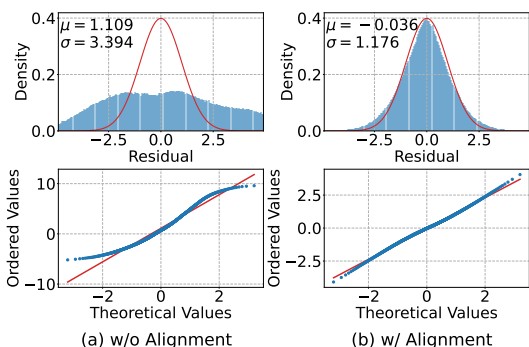

Figure 4: Distribution and QQ-plot of residual noise $\epsilon$ in Eq. (11) before and after representation alignment. Top: red line is the standard Gaussian distribution; blue histogram is the measured distribution. Bottom: red line is the theoretical quantile line; blue dots are the measured values.

sampling ratio $s$ and codebook size $V$, the resulting bit-rate is $bpp = \log_2 V/64s^2$. For decoding, the back-end module of the hyper-encoder upsamples $\mathbf{z}_q$ and projects it back to the original latent space, yielding the quantized latent feature $\tilde{\mathbf{z}} \in \mathbb{R}^{4 \times h \times w}$. The back-end module employs attention layers and additional residual blocks to enhance reconstruction quality. We provide the detailed architecture, along with the training and inference algorithms, in the Appendix.

**Quantized latent representation alignment.** The goal of the first training stage is to align the quantized latent features with the original latent representations. Since downsampling and quantization in the hyper-encoder inevitably discard fine-grained details, we instead focus on preserving structural information by aligning the direction of latent vectors. To this end, we train hyper-encoders using cosine similarity as the quantized latent feature alignment loss:

$$\mathcal{L}_{\mathrm{sim}}^r(\phi) := -\mathbb{E}_{\mathbf{z}_0} \left[ \frac{1}{N} \sum_{n=1}^{N} \mathrm{sim}\left(\mathbf{z}_0^{(n)}, \mathcal{Q}_\phi(\mathbf{z}_0^{(n)}; r)\right) \right], \tag{6}$$

where N is the number of vectors in $\mathbf{z}_0$ and $r$ is the desired compression bit-rate. As in VQ-VAE [51], since vector-quantization (VQ) operation is non-differentiable, the VQ loss is defined as:

$$\mathcal{L}_{\mathrm{VQ}}(\phi) := \mathbb{E}_{\mathbf{z}_0} \left[ \| \mathrm{sg}(\mathbf{z}_0) - \mathbf{z}_q \|_2^2 + \| \mathbf{z}_0 - \mathbf{z}_q \|_2^2 \right], \tag{7}$$

where sg is the stop-gradient operation, and $\mathbf{z}_q$ is the mapping of $\mathbf{z}_0$ to its nearest codebook entry. Following the method in [17], we adopt an exponential moving average (EMA) update for the

codebook, which enhances training stability and eliminates the need for the first term in Eq. (11). Thus, the overall representation alignment objective is:

$$\mathcal{L}_{\text{repa}}^{r}(\phi) := \mathcal{L}_{\text{sim}}^{r}(\phi) + \mathcal{L}_{\text{VQ}}(\phi). \tag{8}$$

We argue that representation alignment is critical for modeling compressed latents as intermediate diffusion states (We will elaborate on this in Section 3.4). To support this, we compare the measured additive noise before and after training. As shown in Fig. 4, prior to training, the residual noise exhibits no clear structure and resembles a uniform distribution within the range of $-2.5$ to $2.5$. The corresponding quantile-quantile plots (QQ-plots) [56] deviate substantially from the theoretical Gaussian line. This indicates that the latents prior to alignment contain only unstructured noise. In contrast, after training, the residual noise closely follows a Gaussian distribution, and the QQ-plots align well with the theoretical line, indicating that the residuals become statistically well-behaved.

### 3.4 Stage 2: Unified Training for One-Step Reconstruction

**Bridging quantization and forward diffusion.** Once the hyper-encoders have stabilized, we empirically observe that the cosine similarity between $\tilde{\mathbf{z}}$ and $\mathbf{z}_0$ remains stable throughout the remainder of training (Please refer to Appendix). This allows us to establish an empirical mapping function between compression bit-rate $r$ and cosine similarity.

$$\hat{\mathrm{F}}_{\text{sim}}^{\phi}(r) := \mathbb{E}_{\mathbf{z}_0}\left[\frac{1}{N}\sum_{n=1}^{N}\text{sim}\left(\mathbf{z}_0^{(n)}, \mathcal{Q}_\phi\left(\mathbf{z}_0^{(n)}; r\right)\right)\right] \tag{9}$$

$$\approx \frac{1}{|\mathcal{D}|N}\sum_{\mathbf{z}_0 \sim \mathcal{D}}\sum_{n}\text{sim}\left(\mathbf{z}_0^{(n)}, \mathcal{Q}_\phi\left(\mathbf{z}_0^{(n)}; r\right)\right), \tag{10}$$

where $\mathcal{D}$ is the training set. We hypothesize that the quantized latent $\tilde{\mathbf{z}}$ corresponds to a diffusion state along a trajectory, where each bit-rate can be associated with a specific diffusion timestep. Formally, we posit that $\tilde{\mathbf{z}}$ can be decomposed as:

$$\tilde{\mathbf{z}} = \sqrt{\bar{\alpha}_t}\mathbf{z}_0 + \sqrt{1 - \bar{\alpha}_t}\,\boldsymbol{\epsilon}, \tag{11}$$

where $\bar{\alpha}_t \in [0, 1]$ is the pseudo diffusion coefficient determined solely by the compression bit-rate, and $\hat{\boldsymbol{\epsilon}} \sim \mathcal{N}(0, \mathbf{I})$ is assumed to be orthogonal to $\mathbf{z}_0$. Under this assumption, the expected cosine similarity between $\tilde{\mathbf{z}}$ and $\mathbf{z}_0$ is:

$$\mathrm{G}_{\text{sim}}(t) := \mathbb{E}_{\hat{\boldsymbol{\epsilon}} \sim \mathcal{N}(0, \mathbf{I})}\left[\text{sim}\left(\sqrt{\bar{\alpha}_t}\mathbf{z}_0 + \sqrt{1 - \bar{\alpha}_t}\,\hat{\boldsymbol{\epsilon}}, \mathbf{z}_0\right)\right] \approx \sqrt{\bar{\alpha}_t}. \tag{12}$$

By matching the empirically measured cosine similarities to this theoretical form, we can infer the corresponding pseudo diffusion timestep $t$ for each bit-rate $r$:

$$\hat{\mathrm{f}}_{\mathcal{R} \to t}^{\phi}(r) := \text{argmin}_{t \in \{1, \cdots, T\}}|\mathrm{G}_{sim}(t) - \hat{\mathrm{F}}_{\text{sim}}^{\phi}(r)|, \tag{13}$$

where $T$ is the maximum diffusion step, $\mathcal{R}$ is the predefined bit-rate set. Figure 4 provides strong empirical support for our modeling. With an appropriate pseudo diffusion timestep $t$, $\hat{\boldsymbol{\epsilon}} = (\tilde{\mathbf{z}} - \sqrt{\bar{\alpha}_t}\mathbf{z}_0)/\sqrt{1 - \bar{\alpha}_t}$ closely approximates the Gaussian distribution.

**Unified fine-tuning across multiple bit-rates.** Remarkably, even under highly compressed conditions, the cosine similarity between $\tilde{\mathbf{z}}$ and $\mathbf{z}_0$ consistently stabilizes at a high level after the first-stage training (Please refer to Appendix). This indicates that the quantized latents still preserve rich structural information, making single-step reconstruction feasible. Leveraging the mapping function $\hat{\mathrm{f}}_{\mathcal{R} \to t}^{\phi}(\cdot)$, we assign each bit-rate a corresponding pseudo diffusion timestep, enabling a shared diffusion backbone to support reconstruction across all bit-rates.

Specifically, given the predefined bit-rate set $\mathcal{R}$, we fine-tune all the corresponding hyper-encoders and apply LoRA [25] to update the diffusion model. The overall loss combines a representation alignment loss $\mathcal{L}_{\text{repa}}^{r}$, a perceptual loss $\mathcal{L}_{\text{per}}$, and an adversarial loss $\mathcal{L}_{\mathcal{G}}$:

$$\mathcal{L}(\phi, \theta) := \mathbb{E}_{r \sim \mathcal{R}}\left[\mathcal{L}_{\text{repa}}^{r}(\phi) + \lambda_1\mathcal{L}_{\text{per}}(\phi, \theta) + \lambda_2\mathcal{L}_{\mathcal{G}}(\phi, \theta)\right]. \tag{14}$$

The alignment loss $\mathcal{L}_{\text{repa}}$ preserves structural consistency between the quantized and original latents. The perceptual loss $\mathcal{L}_{\text{per}}$ includes MSE and DISTS [16] terms. Following the practice of [15], we additionally use the Sobel operator to extract the edge of the image and measure the corresponding edge-aware DISTS loss:

$$\mathcal{L}_{\text{per}}(\phi, \theta) := \mathbb{E}_{\mathbf{I} \sim \mathcal{D}}\left[\mathcal{L}_2(\mathbf{I}, \hat{\mathbf{I}}) + \mathcal{L}_{\text{DISTS}}(\mathbf{I}, \hat{\mathbf{I}}) + \mathcal{L}_{\text{EA-DISTS}}(\mathcal{S}(\mathbf{I}), \mathcal{S}(\hat{\mathbf{I}}))\right], \tag{15}$$

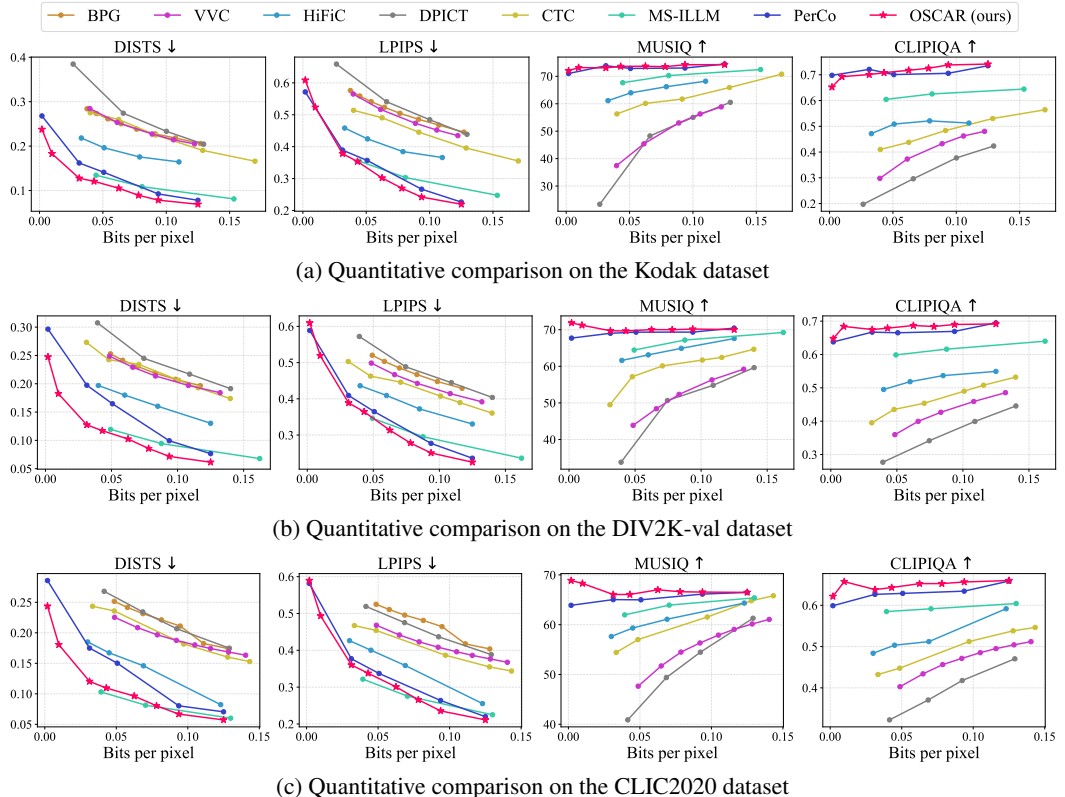

(a) Quantitative comparison on the Kodak dataset

(b) Quantitative comparison on the DIV2K-val dataset

(c) Quantitative comparison on the CLIC2020 dataset

Figure 5: Evaluation of OSCAR and other compression codecs on Kodak, DIV2K-Val, and CLIC2020.

where $\hat{\mathbf{I}} = \mathcal{G}_{\phi,\theta}(\mathbf{z}_0)$, $\mathcal{S}(\cdot)$ is the Sobel operator. Finally, we introduce a discriminator $\mathcal{D}_\psi$ and adopt a GAN loss [3] in latent space to encourage realism of the reconstructed image:

$$\mathcal{L}_{\mathcal{G}}(\phi,\theta) := -\mathbb{E}\left[\log \mathcal{D}_\psi(\hat{\mathbf{z}}_0)\right], \quad \mathcal{L}_{\mathcal{D}}(\psi) := -\mathbb{E}\left[\log(1 - \mathcal{D}_\psi(\hat{\mathbf{z}}_0))\right] - \mathbb{E}\left[\log \mathcal{D}_\psi(\hat{\mathbf{z}}_0)\right], \quad (16)$$

where $\hat{\mathbf{z}}_0 = \mathcal{DN}_\theta(\mathcal{Q}_\phi(\mathbf{z}_0; r), \hat{\mathrm{f}}^\phi_{\mathcal{R} \to t}(r))$ is the reconstructed latent representation.

# 4 Experiments

## 4.1 Experimental Settings

**Datasets and evaluation metrics.** We curate the training data from DF2K, which combines 800 images from DIV2K [2], 2,650 from Flickr2K [48], and an additional subset from LSDIR [33], resulting in 88,441 high-quality images in total. For evaluation, we benchmark OSCAR on three standard datasets: Kodak [18] (24 natural images, 768×512), DIV2K-val [2] (100 images), and CLIC2020 [49] (428 images), where all DIV2K-val and CLIC2020 images are center-cropped to 1024×1024. We adopt both full- and no-reference metrics, including LPIPS [60], DISTS [16], MUSIQ [27], and CLIPIQA [55], to comprehensively assess perceptual and subjective quality.

**Implementation details.** Our approach builds upon Stable Diffusion 2.1 [42], which comprises approximately 965.9M parameters. During the first training phase (Section 3.3), we optimize all hyper-encoders in parallel using the Adam optimizer [28], with a fixed learning rate of $2 \times 10^{-5}$ and a batch size of 64. Training is conducted for 10,000 iterations on a single NVIDIA RTX A6000 GPU. The training converges stably, producing generalized representations across compression levels.

In the second stage, we train our model with the AdamW optimizer [35], setting the learning rate to $5 \times 10^{-5}$, weight decay to $1 \times 10^{-5}$, and batch size to 64 for both OSCAR and the discriminator. We apply LoRA [25] with a rank of 16 for efficient fine-tuning. The discriminator follows the same training protocol as OSCAR. The perceptual loss weight $\lambda_1$ is set to 1, and the adversarial loss weight $\lambda_2$ is $5 \times 10^{-3}$. This stage is trained for 150,000 iterations using eight NVIDIA RTX A6000 GPUs.

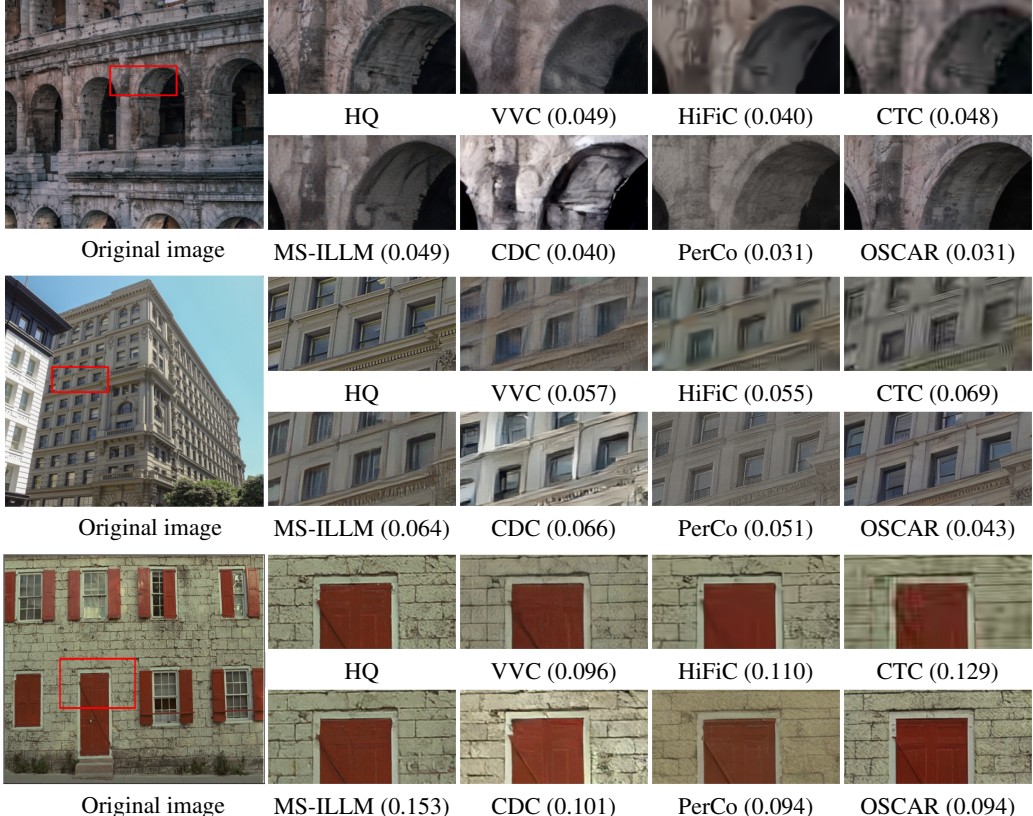

Figure 6: Visual comparison across different bit-rates. HQ denotes the original high-quality patch. The number in parentheses indicates the bpp used for compression. OSCAR yields clean reconstructions while preserving intricate structural and textural details.

## 4.2 Main Results

**Compared methods.** We benchmark OSCAR against four representative categories of image codecs: **1) Traditional methods:** BPG [7] and VVC [11], which serve as hand-crafted baselines. **2) Learning-based single-rate codecs:** HiFiC [37] uses conditional GANs to balance rate, distortion, and perceptual quality, while MS-ILLM [39] enhances local realism via spatially-aware adversarial training. **3) Learning-based multi-rate codecs:** DPICT [31] and CTC [26] both adopt tri-plane coding for flexible rate control. **4) Diffusion-based codecs:** CDC [58] and DiffEIC [34] are optimized for the rate-distortion objective, while PerCo [12] adopts vector quantization. Both DiffEIC and PerCo build upon the same pretrained Stable Diffusion 2.1 model as ours. We retrain PerCo using the open source implementation PerCo (SD) [29]. Each requires a separate model per bit-rate. For HiFiC and CDC, we additionally retrain them on our dataset at lower bit-rates.

**Quantitative results.** The quantitative comparisons on the Kodak, DIV2K-Val, and CLIC2020 datasets are summarized in Fig. 5. OSCAR consistently outperforms all listed baselines across a range of evaluation metrics and compression rates. Among multi-step diffusion methods, we include PerCo, which also employs a codebook quantization strategy similar to ours. Further comparisons with other multi-step baselines are provided in the appendix. OSCAR's strong results on perceptual metrics such as LPIPS and DISTS highlight its ability to preserve fine details and achieve high perceptual fidelity, while its performance on no-reference metrics like CLIPIQA further confirms the visual quality of the reconstructions.

**Discussion of the trend of RD-curves.** The RD-curves of OSCAR are not strictly convex, as the bitrate is jointly determined by both the downsampling ratio of the hyper-encoder and the size of the codebook. When the bitrate changes, the effects introduced by these two factors may not vary consistently. Nevertheless, our method maintains smooth transitions between adjacent bitrates without noticeable performance drops, while still achieving strong results at low bitrates.

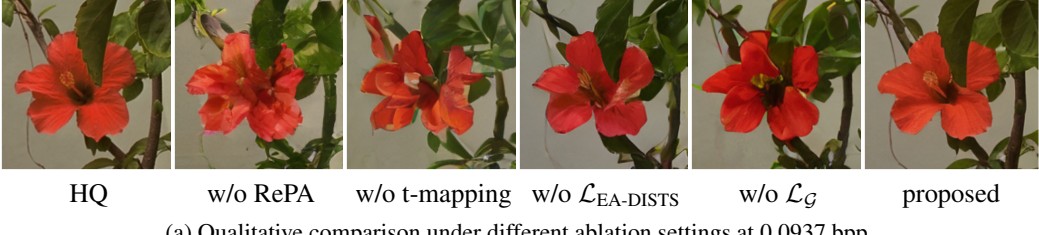

HQ  w/o RePA  w/o t-mapping  w/o $\mathcal{L}_{\text{EA-DISTS}}$  w/o $\mathcal{L}_{\mathcal{G}}$  proposed

(a) Qualitative comparison under different ablation settings at 0.0937 bpp.

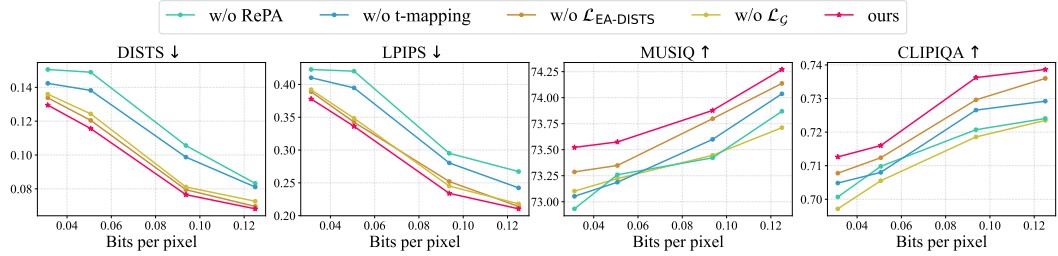

(b) Quantitative results on the Kodak dataset under different ablation settings.

Figure 7: Ablation study on key components. RePA denotes representation alignment, and w/o t-mapping refers to using random mapping between bit-rates and time-steps.

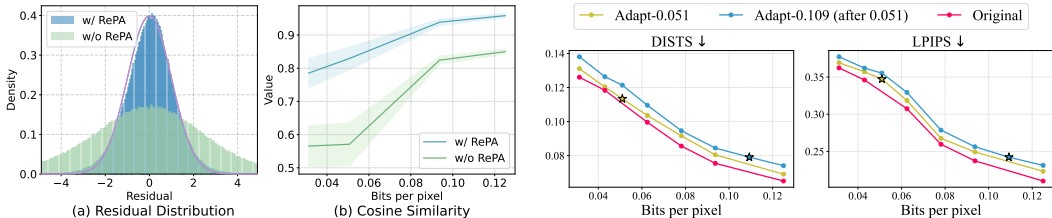

(a) Effect of representation alignment. The purple curve in the left plot is the standard Gaussian.

(b) Results of generalization to unseen bitrates. The star denotes the newly adapted bitrate.

Figure 8: Ablation study on RePA and generalization to unseen bitrates.

**Qualitative results.** As shown in Fig. 6, traditional, learning-based, and diffusion-based codecs generally preserve the coarse structure of the image but suffer from noticeable quality degradation, such as blurring, texture loss, color shifts, and visible artifacts. For instance, wall textures often appear overly smooth, while fine structures such as window frames, door edges, and decorative patterns are frequently distorted or lost. In contrast, OSCAR preserves such details with high fidelity, maintaining both material realism and structural consistency with the original image. Overall, it achieves a highly favorable rate-distortion trade-off, ensuring that compressed outputs retain both excellent perceptual quality and faithful structural integrity. We provide more results in the appendix.

### 4.3 Ablation Studies

**Mapping bit-rates to pseudo diffusion timesteps.** In our method, each compression bit-rate is associated with a pseudo diffusion timestep through a learned mapping function. To evaluate its effectiveness, we compare this design to a baseline that randomly assigns timesteps. As shown in Fig. 7a, removing the calibrated mapping leads to noticeable visual degradation—reconstructed images fail to preserve the original shape and structure. This degradation is further reflected in Fig. 7b, which shows consistent performance drops across all evaluation metrics and bit-rates.

**Representation alignment.** To validate the effectiveness of representation alignment, we compare the residual term in Eq. (11) with and without alignment. As shown in Fig. 8a, the aligned model produces residuals that closely match the standard Gaussian, whereas the unaligned one shows large deviations. Cosine similarity between quantized and original latents is also notably higher with

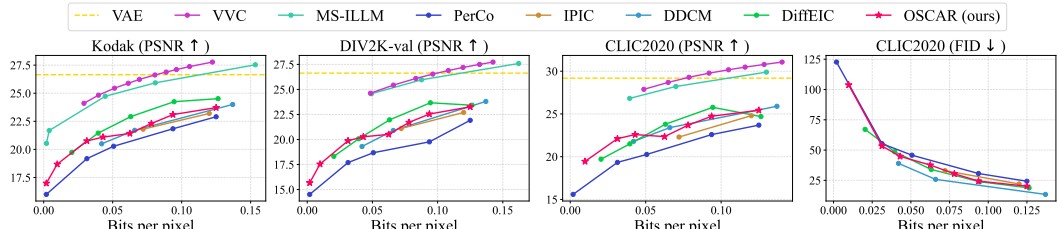

Figure 9: PSNR and FID results on Kodak, DIV2K-val, and CLIC2020 datasets. The yellow dashed line indicates reconstruction by VAE encoding and decoding without compression.

alignment training. Figure 7b further demonstrates the effectiveness of alignment through both visual and quantitative comparisons, showing that removing it leads to substantial performance degradation.

**OSCAR training loss functions.** OSCAR incorporates multiple loss functions during training to enhance performance. To assess the impact of key components, we perform ablation studies on the edge-aware DISTS loss and GAN loss. As shown in Fig. 7a, removing either loss leads to a noticeable reduction in visual detail. Figure 7b further shows performance degradation, particularly on non-reference metrics, indicating their importance for perceptual quality.

**Generalization to unseen bitrates.** For each predefined bitrate, OSCAR employs an individual hyper-encoder for compression, resulting in a set of fixed-rate models. Nevertheless, we demonstrate that OSCAR exhibits strong generalization capability to unseen bitrates. Specifically, for a new bitrate, we continue training for 15k iterations under the same settings as before, except that in each iteration, the bitrate is sampled from the new one with a probability of 0.5 and from the existing set with a probability of 0.5. As shown in Fig. 8b, the new bitrate can be effectively learned with only minor performance degradation on the previously trained bitrates.

**PSNR and FID results.** We present the PSNR and FID performance of state-of-the-art codecs in Fig. 9. In addition to the compared baselines, we further include IPIC [57] and DDCM [40], whose results on PSNR and FID are highly competitive. For FID evaluation, we report results only on the CLIC2020 dataset, as the other two datasets contain too few images for statistically stable estimation.

Diffusion-based codecs generally underperform traditional methods in terms of PSNR. This is mainly due to the inherent limitation of VAEs, which struggle to achieve pixel-wise accurate reconstruction. As indicated by the yellow dashed line in the figure, the reconstruction from a VAE (without compression) can even yield lower PSNR than traditional codecs at higher bitrates. Despite these challenges, OSCAR demonstrates competitive performance among multi-step diffusion-based codecs.

## 5    Conclusion

We propose OSCAR, a one-step diffusion codec that supports compression across multiple bit-rates within a unified network. By modeling quantized latents as intermediate diffusion states and mapping bit-rates to pseudo diffusion timesteps, OSCAR achieves efficient single-pass reconstruction through a shared generative backbone. This work establishes a novel pathway for efficient, practical, and high-fidelity diffusion-based image compression in real-world applications. Extensive experiments demonstrate the superiority of OSCAR over recent leading methods.

## Acknowledgments

This work was supported by Shanghai Municipal Science and Technology Major Project (2021SHZDZX0102) and the Fundamental Research Funds for the Central Universities.

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
