# OpenReview forum: "OSCAR: One-Step Diffusion Codec Across Multiple Bit-rates"
_NeurIPS.cc/2025/Conference — NeurIPS 2025 poster_

### Official Review · Reviewer_7kzL · 2025-06-09

**Clarity:** 2
**Significance:** 4
**Originality:** 4
**Rating:** 5
**Confidence:** 5

**Summary:**

The paper introduces OSCAR, a one-step diffusion-based compression method that supports multiple bit-rates using a single generative model. This is different from previous diffusion based works that require iterative denoising, and usually need multiple models for multiple timesteps. They do this by basically changing the forward process to be the compression (quantized latent vectors), such that the reverse process yield decompression. This is quite different from previous diffusion-based approaches that leverage the denoising prior directly. The qualitative results, while lacking, show promising performance.

**Questions:**

- How many parameters are added by the rank 16 LoRAs?
- If PerCo (SD) was used, they release just 3 checkpoints, 0.1250bpp, 0.0313bpp and 0.0019bpp, yet the authors include also for 0.051bpp and 0.094bpp (Fig. 6). Did the authors finetune those themselves? If so, this should be noted.

**Ethical Concerns:**

["NO or VERY MINOR ethics concerns only"]

**Final Justification:**

All of my concerns have been addressed:
- The authors promised to fix the style issue - which is very important to me.
- They compared in the rebuttal to missing related works and promised to include those results and discuss further related work.
- The included PSNR and FID here and say they will include them in the paper.
- They clarified details such as the used PerCo version.
- They promised to expand the limitation section.
- They will include further visualizations.

**Limitations:**

Somewhat. The authors note in the supplementary how the method supports only a small amount of bit-rates at once, yet they do not relate how the pre-determined choice of bitrates requires retraining the entire model when support should be added for a single new bitrate. Additionally, they do not discuss how the latent-space nature of the method hurts pixel-space based distortion.

**Paper Formatting Concerns:**

The authors **played with the template’s caption, section and paragraph spacings** in such a way that left the reading very confusing to this reviewer at times. For example, the captions of Fig. 1 and 8 merged with the content paragraph and made for an uneasy reading. Sec 4.2 was squeezed such that paragraphs had no room to breathe. This is quite problematic - as it is not fair towards other papers that did shorten their paper to fit neatly into the page limit.

**Quality:**

4

**Strengths And Weaknesses:**

## Strengths
- The proposed idea of changing the “degradation” to be the compression is an interesting and novel idea. Specifically, the choice of a quantization degradation which can be modeled as additive noise is elegant.
- Single model for multiple bitrates.
- One-step sampling is great and efficient.
- The ablation studies are helpful and support the authors’ claims.
- The algorithms provided in the appendix are helpful.
## Weaknesses
- The authors **played with the template’s caption, section and paragraph spacings** in such a way that left the reading very confusing to this reviewer at times. For example, the captions of Fig. 1 and 8 merged with the content paragraph and made for an uneasy reading. Sec 4.2 was squeezed such that paragraphs had no room to breathe. This is quite problematic – as it is not fair towards other papers that did shorten their paper to fit neatly into the page limit.
- **Missing (highly) related works**:
  - L32 – the original paper of Sohl-Dickstein 2015 should be cited.
  - A very relevant missing work is the DDCM compression approach [1], which uses the entire diffusion process for compression and complementary exact restoration. It shares similarity with the current paper, since they also use the gaussian noise to represent the intermediate compression vectors, which can be seen as using the original DDPM degradation directly – in a zero-shot manner – to provide similar effects. As a result, they also do not need multiple models for different bitrates (L40). This should be cited and compared to accordingly. While it suffers from the same multi-sampling problem, it could also serve as a non-bitrate-specific diffusion-based baseline which the paper is missing.
  - Another missing related work is with regards to diffusion models that change the forward model. For example, Cold Diffusion [2] propose a pixelated degradation version that is conceptually similar to what is proposed here and should be explicitly compared to.
- The paper includes multiple comparisons to PerCo. However, their internal model was not released. Did the authors train their own PerCo version, or used the open-sourced version, "PerCo (SD)"? If the latter, the corresponding paper [3] should be cited.
- **Missing important metrics:** PSNR and FID metrics are missing from the evaluations.
  - Specifically, in this reviewers experience, MUSIQ and especially CLIPIQA are not corelative with compression performance on visual inspection, and FID is commonplace to report, e.g. [1,3].
  - PSNR is widely used (in addition to the MS-SSIM reported in the appendix) to allow for consistent fair comparisons. At least one such pixel-space metric, preferably PSNR, should be in the main text, to showcase the inherent limitation in using latent-space-based methods (as mentioned in L121 in the appendix). This limitation should be discussed and noted as such.
  - Even when considering the reported MS-SSIM metric in Fig. 3, for some reason the authors omit BPG, CTC and DPICT. This is problematic, as this reviewer would expect this methods to perform well in pixel-space distortion, especially in higher bitrates.
- As the authors note in the supplementary, the method supports multiple bit-rates with one model, however it is limited to a small set of pre-chosen bitrates. This is unlike variable rate codecs (e.g. BPG or [1]). The authors mention how learning a single model for multiple bitrates is hard. An additional weakness is the need to retrain the entire model for every new wanted bitrate that would be added later, unlike e.g. PerCo which would just finetune for the new bitrate a pre-trained encoder that has most of the prior intact.
- The visualization choice to only show small crops from the images is problematic in this reviewer’s opinion. The authors should include full images comparisons, or at least large crops, to see how the methods affects the entire image and not just a precisely selected small patch.

[1] Ohayon, Guy, Hila Manor, Tomer Michaeli, and Michael Elad. "Compressed Image Generation with Denoising Diffusion Codebook Models." arXiv preprint arXiv:2502.01189 (2025).

[2] Yen, Hao, François G. Germain, Gordon Wichern, and Jonathan Le Roux. "Cold diffusion for speech enhancement." In ICASSP 2023-2023 IEEE International Conference on Acoustics, Speech and Signal Processing (ICASSP), pp. 1-5. IEEE, 2023.

[3] Körber, Nikolai, Eduard Kromer, Andreas Siebert, Sascha Hauke, Daniel Mueller-Gritschneder, and Björn Schuller. "PerCo (SD): Open Perceptual Compression." arXiv preprint arXiv:2409.20255 (2024)

---

> ### Author Rebuttal · Authors · 2025-07-30
>
> Thanks for the thorough review and valuable comments. We are encouraged by your recognition of the elegance of our method and the clarity of our empirical results. Below we respond to your specific comments.
>
> - Q1: The paragraph spacing issue.
>
>     Thank you very much for your attention to this matter. We strictly followed the official LaTeX template, but acknowledge that some section and caption placements may have affected readability. We will carefully revise the formatting in our revised paper to ensure a clear presentation.
>
> - Q2: Authors should expand the related work.
>
>     Thanks for the suggestion. We will add the citations and include the comparison results in the revised paper, as well as a discussion of similar works in the related work section.
>
>     - Comparison with DDCM compression approach.
>
>         In our method, we use LPIPS-AlexNet as a training loss, and LPIPS-VGG as the evaluation metric to ensure a fair comparison. Here we follow DDCM to use LPIPS-AlexNet (denoted as LPIPS*) and crop DIV2K-val images to 768$\times$768 for evaluation.
>
>         | model | bpp | PSNR↑ | LPIPS*↓ | FID↓ |
>         | --- | --- | --- | --- | --- |
>         | DDCM (s=1000) | 0.0420 | 21.1670 | 0.2010 | 18.3300 |
>         | OSCAR (s=1) | 0.0313 | 21.3304 | 0.1958 | 31.8673 |
>         | DDCM (s=1000) | 0.0660 | 21.7800 | 0.1720 | 16.3960 |
>         | OSCAR (s=1) | 0.0507 | 21.9478 | 0.1554 | 28.9917 |
>         | DDCM (s=1000) | 0.1370 | 22.7070 | 0.1380 | 14.6620 |
>         | OSCAR (s=1) | 0.1250 | 23.4529 | 0.1097 | 15.7407 |
>
>         Compared to the 1000-step DDCM, our one-step OSCAR achieves better performance on reference-based metrics (PSNR and LPIPS*), indicating more accurate reconstructions. Although DDCM yields better FID scores, the performance gap narrows as the bitrate increases.
>
>     - Comparison with Cold Diffusion.
>
>         We carefully reviewed this paper and found that it focuses on speech enhancement, without an image compression module and an open-source implementation. We'd appreciate it if you could clarify how it might be adapted to image compression.
>
>     - What is the PerCo Version? Did the authors finetune 0.051 bpp and 0.094 bpp PerCo themselves?
>
>         We use PerCo (SD) as our baseline. For 0.0313 bpp and 0.1250 bpp, we use the checkpoints released in their open-source code. For 0.051 bpp and 0.094 bpp, we train the model ourselves using their codebase. We will clarify this in the revised manuscript and include the corresponding paper in our citations.
>
> - Q3: Authors should supplement PSNR, MS-SSIM, and FID results, and discuss how the latent-space nature of the method hurts pixel-space distortion.
>
>     Thank you for the valuable advice.  We omitted the MS-SSIM results of BPG, DPICT, and CTC in the original paper as they generally underperform MS-ILLM, which we reported as a representative non-diffusion baseline. We will include the full results and the detailed discussion in our revised paper.
>
>     Tables below present results on the DIV2K-val dataset.
>
>     | model | bpp | PSNR↑ | MS‑SSIM↑ | FID↓ |
>     | --- | --- | --- | --- | --- |
>     | BPG | 0.0584 | 23.9356 | 0.8624 | 127.6502 |
>     | DPICT | 0.0745 | 24.5078 | 0.8896 | 78.0774 |
>     | CTC | 0.0707 | 25.8485 | 0.9166 | 75.4264 |
>     | MS‑ILLM | 0.0493 | 24.5745 | 0.8814 | 42.6882 |
>     | PerCo (s=20) | 0.0507 | 19.0644 | 0.6883 | 29.8819 |
>     | DiffEIC (s=50) | 0.0414 | 21.7711 | 0.7586 | 29.4854 |
>     | OSCAR (s=1) | 0.0507 | 21.9662 | 0.7623 | 28.8701 |
>
>     | model | bpp | PSNR↑ | MS‑SSIM↑ | FID↓ |
>     | --- | --- | --- | --- | --- |
>     | BPG | 0.1361 | 26.1172 | 0.9293 | 94.8704 |
>     | DPICT | 0.1401 | 26.3581 | 0.9336 | 41.4491 |
>     | CTC | 0.1399 | 26.8273 | 0.9365 | 35.2881 |
>     | MS‑ILLM | 0.1622 | 27.0104 | 0.9417 | 18.3835 |
>     | PerCo (s=20) | 0.1250 | 21.9182 | 0.8613 | 16.8871 |
>     | DiffEIC (s=50) | 0.1264 | 23.4237 | 0.9088 | 15.1083 |
>     | OSCAR (s=1) | 0.1250 | 23.4810 | 0.8906 | 15.7285 |
>
>     We observe that diffusion codecs generally outperform non-diffusion methods in perceptual metrics such as FID (shown in the table), LPIPS, and DISTS (reported in our main paper), but underperform in pixel-level fidelity metrics like PSNR and SSIM. OSCAR, along with PerCo, DiffEIC, and the recent DDCM, all exhibit this behavior. We attribute this common behavior to the inherent characteristics of the diffusion backbone.
>
>     This trade-off between pixel-wise fidelity and perceptual quality arises mainly from two factors. (1) The strong generative prior in pre-trained diffusion models, which prioritizes perceptual quality over pixel-wise reconstruction. (2) The nature of denoising in latent space amplifies this trade-off, as the decoder must reconstruct images from compressed, high-level features, which are optimized for semantic or perceptual quality rather than pixel accuracy. Unlike pixel-space compression, latent-space representations inherently discard low-level details, making exact pixel-wise reconstruction more difficult.
>
>     Notably, this gap in pixel-level fidelity tends to narrow at higher bitrates, especially for MS-SSIM. This suggests that the limitations of latent-space reconstruction can be gradually mitigated when more information is retained.
>
> - Q4: The method supports multiple bitrates with one model, but is limited to pre-chosen bitrates.
>
>     The goal of our method is not to support a very fine-grained range of bitrates, as in traditional variable-bitrate approaches. Rather, we aim to explore whether the proposed bitrate-to-timestep mapping can enable a one-step diffusion model to support a set of bitrates while achieving performance comparable to single-bitrate models. Our experiments provide strong evidence that OSCAR is effective in achieving this target.
>
>     To the best of our knowledge, current variable-bitrate methods still underperform single-bitrate baselines (Figure 5 in our paper). While our approach is currently validated on a set of predefined bitrates, we believe it offers a promising path toward achieving both flexibility and high performance in future designs.
>
> - Q5: The entire model needs to be retrained for every new wanted bitrates, while PerCo just finetunes a pre-trained encoder.
>
>     We acknowledge that the diffusion backbone needs to be retrained for new bitrates. However, we demonstrate that OSCAR can generalize well to new bitrates with minimal computational overhead.
>
>     We adopted a simple training scheduler: half of the training steps are allocated to the new bitrate, and the other half to the original bitrate group. Extending to a new bitrate requires only 15K training iterations, which take less than 4 hours on 4×A6000 GPUs. We attribute this efficiency to the fact that the diffusion model has already learned to handle structurally similar latent distributions, enabling fast and stable adaptation to nearby bitrates, while causing minimal performance degradation on previously trained bitrates.
>
>     The tables below report results on the DIV2K-val dataset, where we progressively adapt the model to new bitrates one at a time. Numbers in parentheses indicate the performance change compared to the model before adaptation to the new bitrate. These results will be included in the revised paper.
>
>     | bpp | DISTS↓ | LPIPS↓ | MS-SSIM↑ |
>     | --- | --- | --- | --- |
>     | 0.0313 | 0.1262 | 0.3883 | 0.7210 |
>     | 0.0507 | 0.1102 | 0.3487 | 0.7623 |
>     | 0.0937 | 0.0694 | 0.2385 | 0.8713 |
>     | 0.1250 | 0.0617 | 0.2157 | 0.8906 |
>
>     | bpp | DISTS↓ | LPIPS↓ | MS-SSIM↑ |
>     | --- | --- | --- | --- |
>     | 0.0313 | 0.1306 (↑3.49%) | 0.3927 (↑1.13%) | 0.7161 (↓0.68%) |
>     | 0.0507 | 0.1141 (↑3.54%) | 0.3523 (↑1.04%) | 0.7578 (↓0.59%) |
>     | 0.0937 | 0.0728 (↑4.90%) | 0.2427 (↑1.77%) | 0.8595 (↓1.35%) |
>     | 0.1250 | 0.0648 (↑4.93%) | 0.2198 (↑1.91%) | 0.8766 (↓1.57%) |
>     | **0.0430 (new)** | 0.1198 | 0.3670 | 0.7402 |
>
>     | bpp | DISTS↓ | LPIPS↓ | MS-SSIM↑ |
>     | --- | --- | --- | --- |
>     | 0.0313 | 0.1360 (↑4.19%) | 0.4006 (↑2.03%) | 0.7031 (↓1.81%) |
>     | 0.0507 | 0.1185 (↑3.87%) | 0.3589 (↑1.87%) | 0.7459 (↓1.57%) |
>     | 0.0937 | 0.0752 (↑3.34%) | 0.2457 (↑1.25%) | 0.8506 (↓1.03%) |
>     | 0.1250 | 0.0672 (↑3.75%) | 0.2228 (↑1.38%) | 0.8664 (↓1.16%) |
>     | 0.0430 | 0.1246 (↑4.01%) | 0.3742 (↑1.98%) | 0.7282 (↓1.62%) |
>     | **0.0781 (new)** | 0.0918 | 0.2766 | 0.8412 |
>
>     In addition, we carefully examined the training procedure of PerCo. In their original paper, they state that all linear layers in the UNet (about 15% of the model) must be finetuned for each new bitrate. In contrast, our method uses LoRA to finetune only 1.8% of the total parameters. PerCo (SD) finetunes the entire UNet and hyper-encoder according to the open-source implementation. We kindly request you to double-check this point.
>
> - Q6: The visualization should include full images or large crop comparisons.
>
>     Thank you for the suggestion. We have prepared the full image comparisons. However, we noticed that additional PDFs are not permitted in rebuttal this year. We will include these visual results in the revised version of our paper.
>
> - Q7: How many parameters are added by the rank 16 LoRA?
>
>     The SD-2.1 UNet has 882.1M parameters, and the rank-16 LoRA introduces an additional 16.2M parameters.
>
>
> ---
>
> We hope this response can help address your concerns. We believe our work can have an important impact on the field of image compression by offering a new perspective on leveraging pre-trained diffusion models for efficient one-step reconstruction across multiple bitrates. If you find our response satisfactory, we would be sincerely grateful if you could consider a higher score, as it would greatly support our efforts. Thank you once again for your consideration.

---

> > ### Comment · Reviewer_7kzL · 2025-08-01
> >
> > I thank the authors for their thorough response and for providing additional experiments.
> > - I am very sorry - I cited a wrong paper by the same name. I was referring to [Bansal et al. 2022](https://arxiv.org/abs/2208.09392), which [published their code](https://github.com/arpitbansal297/Cold-Diffusion-Models). As you can now see, I was referring to their "pixelate" degradation.
> > - A3: I assume the wide gap in distortion is mainly due to the latent VAE of the diffusion model. I would suggest adding a comparison to just passing the original image through the VAE and explicitly compare to this bound.
> > - A4: I agree, it is still a limitation by design.
> >
> > I am generally leaning towards increasing my score - as long as the authors commit to carefully following the template's spacing in the revised version and appending the discussed matters to their paper.

---

> ### Author Response · Authors · 2025-08-01
>
> Thank you very much for the response. We appreciate your inclination to increase the rating. **We commit to further refining the spacing in line with the template to enhance readability, as well as appending the discussed matters to our revised paper.** Below, we provide our detailed response to the remaining concern.
>
> - Q1: Comparison with Cold Diffusion.
>
>     Thank you for the clarification. Cold Diffusion is specifically designed for image restoration tasks such as deblurring, denoising, and super-resolution. However, based on both the paper and its open-source implementation, we do not find a straightforward way to adapt it for image compression. Nevertheless, the pixelated degradation idea is interesting and inspiring, and we will include this paper in the related work section of our paper.
>
> - Q2: The reconstruction bound for VAE.
>
>     We conducted an additional experiment by directly passing the original image through the SD-2.1 VAE. As shown in the table below, the pixel-wise fidelity of the VAE is relatively low—only comparable to CTC at 0.1399 bpp. This provides a clear explanation for why diffusion models tend to lag behind in pixel-level fidelity. We value this insightful suggestion and will include these results in the revised paper.
>
> | model | Dataset | PSNR↑ | MS‑SSIM↑ |
> | --- | --- | --- | --- |
> | SD-VAE | Kodak | 26.6492 | 0.9218 |
> | OSCAR (0.1250bpp) | Kodak | 23.9742 | 0.8778 |
> | SD-VAE | DIV2K-val | 26.6280 | 0.9431 |
> | OSCAR (0.1250bpp) | DIV2K-val | 23.4810 | 0.8906 |

---

> > ### Comment · Reviewer_7kzL · 2025-08-03
> >
> > Thank you for the response. I am raising my score, in my opinion the paper should be accepted if all that was discussed would be fixed/mentioned.

---

### Official Review · Reviewer_nsjq · 2025-06-22

**Clarity:** 3
**Significance:** 2
**Originality:** 2
**Rating:** 4
**Confidence:** 4

**Summary:**

This paper proposed a one-step diffusion codec across multiple bit-rates called OSCAR. It views compressed latents as noisy variants of the original latents and  the level of distortion depends on the bit-rate. This enables the model to support reconstructions at multiple bit-rates. The compressed latents retain rich structural information, thus making one-step diffusion feasible and improve the inference efficiency.

**Questions:**

- In Figure 6, the paper presents visual comparisons, but the results of DiffEIC are not included. Could you provide visual comparisons with DiffEIC to facilitate a more comprehensive evaluation?
- While the diffusion-based method can generate fine details, it may lead to severe discrepancies from the original image. Do the proposed method show better fidelity than existing diffusion-based method (e.g. DiffEIC)?
- Some existing methods typically introduce quantization parameters to control the bitrate, which is both simple and practical in applications. In contrast, the proposed method introduces a mapping between bitrates and pseudo diffusion timesteps to enable variable bit-rate coding. Could the authors elaborate on the advantages of this design choice? Additionally, is there a reason for not combining this approach with existing methods to leverage the strengths of both?

- The complexity comparison in Figure 2. Could you provide the number of the MACs and running time of OSCAR, MS-ILLM and DiffEIC for better comparion? Moreover, both encoding and decoding time should be provided.
- The RD-curves presented in Figure 7 appear somewhat unusual, as they do not exhibit the typical convex or concave shapes commonly observed in rate-distortion analysis (depending on the metric used). Furthermore, disabling certain key components leads to significantly different trends in the curves, which raises questions about the consistency of the experimental setup. The authors are encouraged to double-check their experimental results and provide additional analysis.
- The proposed method supports variable bit-rate coding by views compressed latents as noisy variants of the original latents. What is the range of bitrates that the proposed model can effectively support? It would be helpful if the authors could clarify the bitrate scalability of the model. Can the model still work well at extremely low-bitrates?

**Ethical Concerns:**

["NO or VERY MINOR ethics concerns only"]

**Final Justification:**

Considering that the authors have addressed my previous concerns well and this paper is one of the early works that introduce one-step diffusion into neural image compression, I raise my score to 4: Borderline accept.

**Limitations:**

yes

**Paper Formatting Concerns:**

I do not notice obvious paper formatting concerns.

**Quality:**

2

**Strengths And Weaknesses:**

Strengths:
- The proposed model use one-step diffusion for compression, leading to improved inference efficiency compared to existing methods.
- The proposed method supports variable bit-rate coding by views compressed latents as noisy variants of the original latents.


Weaknesses:
- Additional experiments and analysis are expected to improve this paper, see the Questions part for more details.
- Visual comparison with DiffEIC in Figure 6 should be included and analysis about the RD-curves presented in Figure 7 should be provided. See **Questions** for more details.

---

> ### Author Rebuttal · Authors · 2025-07-30
>
> Thanks for the valuable comments, constructive advice, and recognition of our idea and empirical results. Below we respond to your specific comments.
>
> - Q1: Visual comparisons with DiffEIC.
>
>     Thank you for the suggestion. We have prepared the visual comparisons with DiffEIC. However, as additional PDFs are not permitted in rebuttal this year, we will include these visual results in the revised version of our paper.
>
> - Q2: Does the proposed method show better fidelity than the existing diffusion-based method?
>
>     Pre-trained diffusion models inherently prioritize perceptual quality over exact pixel-wise reconstruction due to their strong generative priors. As a result, diffusion codecs often produce visually appealing images at the cost of pixel-level fidelity. Since OSCAR also builds upon a diffusion backbone, this trade-off naturally applies. However, despite supporting multiple bitrates and requiring only a single denoising step, OSCAR achieves performance competitive with DiffEIC and significantly surpasses PerCo in both perceptual and pixel-level fidelity.
>
>     The table below reports results on the DIV2K-val dataset:
>
>     | models | bpp | DISTS↓ | LPIPS↓ | MS-SSIM↑ |
>     | --- | --- | --- | --- | --- |
>     | PerCo (s=20) | 0.0507 | 0.1647 | 0.3646 | 0.6883 |
>     | DiffEIC (s=50) | 0.0422 | 0.1346 | 0.3501 | 0.7586 |
>     | OSCAR (s=1) | 0.0507 | 0.1102 | 0.3487 | 0.7623 |
>     | PerCo (s=20) | 0.0937 | 0.0993 | 0.2769 | 0.8081 |
>     | DiffEIC (s=50) | 0.0945 | 0.0687 | 0.2275 | 0.8838 |
>     | OSCAR (s=1) | 0.0937 | 0.0694 | 0.2285 | 0.8773 |
> - Q3: Could the authors elaborate on the advantages of mapping bitrates to diffusion timesteps?
>
>     The core idea of our approach is that for each bitrate, the quantization error can be well approximated as Gaussian noise at a specific diffusion timestep. By assigning the most appropriate timestep to each bitrate, OSCAR can more effectively leverage a pre-trained diffusion model for image compression. This design is validated by our extensive experiments, showing that OSCAR can support multiple bitrates using only one-step diffusion while achieving competitive or even superior performance compared to existing multi-step diffusion models that operate at a single bitrate.
>
>     As for the bitrate control, most existing variable-bitrate methods rely on hyperparameters such as quality factor (e.g., BPG and VVC) and compression level (e.g., CTC). This typically requires users to tune these parameters to match a target bitrate. In contrast, our method allows users to specify the bitrate explicitly, thereby offering a more user-friendly interface for bitrate control.
>
> - Q4: Can we combine this approach with existing variable-bitrate coding?
>
>     As discussed in Q3, our bitrate-to-timestep mapping is specifically designed to model the quantized latent as a diffusion state, enabling better utilization of the pre-trained diffusion model for image compression. This design is fundamentally different from existing variable-bitrate methods that rely on quantization parameters to control compression levels. Therefore, these methods cannot be simply combined.
>
> - Q5: The numerical value of the MACs and running time of the leading baselines should be provided.
>
>     Thank you for the suggestion. We will include the number of MACs and the runtime (covering both encoding and decoding) in our paper.
>
>     The results below are measured on images of size 1024$\times$1024.
>
>     | models | Encoding (ms) | Decoding (ms) | Total (ms) | MACs (T) |
>     | --- | --- | --- | --- | --- |
>     | MS-ILLM | 208.77 | 314.72 | 523.49 | 1.05 |
>     | DiffEIC | 478.70 | 21821.79 | 22300.49 | 304.87 |
>     | OSCAR | 239.48 | 622.26 | 861.74 | 8.54 |
> - Q6: Additional analysis on the RD-curves presented in Figure 7.
>
>     We agree that RD-curves typically exhibit convex or concave trends. However, in our method, the bitrate is jointly determined by the downsampling ratio and codebook size. Specifically, 0.0937 and 0.1250 share the same downsampling ratio, as do 0.0313 and 0.0507. Bitrates with the same downsampling factor tend to show more similar distortion behaviors, leading to the observed patterns in our RD-curve. Appendix Figure 1 further illustrates this, bitrates with the same downsampling ratio also show similar cosine similarity. A similar trend is also observed in PerCo (Fig. 5(a)), which also uses a similar codebook-based design.
>
>     RD-curves typically measure the distortion between reconstructed and original images. For no-reference IQA metrics such as MUSIQ and CLIP-IQA, the usual convex or concave trends may not apply, as these metrics are influenced by perceptual quality beyond simple image similarity. As a result, removing key modules may alter the curve’s shape more significantly. Therefore, while these metrics are valuable for comparing overall perceptual quality, they may be less suitable for analyzing RD curve trends.
>
>     We confirm the consistency of our experimental setup and will release all code and models to ensure reproducibility.
>
> - Q7: The authors should clarify the bitrate scalability of the model.
>
>     Thank you for the helpful comment. We retrained OSCAR to include two lower bitrates and evaluated its performance in the extremely low-bitrate setting. The results show that OSCAR achieves competitive performance down to 0.0098 bpp, demonstrating its scalability across a broad bitrate range (0.0098 to 0.1250). These results will be included in the revised manuscript.
>
>     The table below shows results on the DIV2K-val dataset. Numbers in parentheses indicate relative change compared to the original version of OSCAR.
>
>     | models | bpp | DISTS↓ | LPIPS↓ | MS-SSIM↑ |
>     | --- | --- | --- | --- | --- |
>     | OSCAR | 0.0313 | 0.1274 (↑0.95%) | 0.3866 (↓0.44%) | 0.7234 (0.33%↑) |
>     | OSCAR | 0.0507 | 0.1098 (↓0.36%) | 0.3479 (↓0.23%) | 0.7644 (0.28%↑) |
>     | OSCAR | 0.0937 | 0.0702 (↑1.15%) | 0.2398 (↑0.55%) | 0.8694 (↓0.22%) |
>     | OSCAR | 0.1250 | 0.0630 (↑0.96%) | 0.2170 (↑0.60%) | 0.8881 (↓0.28%) |
>     | OSCAR | **0.0234 (new)** | 0.1494 | 0.4265 | 0.6993 |
>     | PerCo (SD) | 0.0234 | 0.2359 | 0.4433 | 0.6472 |
>     | OSCAR | **0.0098 (new)** | 0.1820 | 0.4628 | 0.6556 |
>     | PerCo (SD) | 0.0098 | 0.2870 | 0.4841 | 0.5889 |
>
> ---
>
> We hope this response can help address your concerns. We believe our work can have an important impact on the field of image compression by offering a new perspective on leveraging pre-trained diffusion models for efficient one-step reconstruction across multiple bitrates. If you find our response satisfactory, we would be sincerely grateful if you could consider a higher score, as it would greatly support our efforts. Thank you once again for your consideration.

---

> ### Comment · Reviewer_nsjq · 2025-08-03
>
> Thanks for your answer, but I think some important concerns are still not solved.
>
> Firstly, I must point out some issues in your answer to Q2:
>
> The original intent of my question was to know whether using only a single-step diffusion step might lead to greater semantic deviation compared to multi-steps. I know that the current perceptual metrics (e.g., LPIPS and DISTS) are limited, where higher scores may not mean better fidelity. That's also the reason that why I ask for the visual comparison with DiffEIC (The visual comparison could visually demonstrate the difference in fidelity).
> I observed that you show the performance of Perco on high-resolution DIV-2k, however, that is not valuable. Perco performs much worse on high resolution images than low-resolution ones due to its design. They target at low-resolution in their paper.
>
> models	          bpp	DISTS↓	LPIPS↓	MS-SSIM↑
>
> DiffEIC (s=50)	0.0422	0.1346	0.3501	0.7586
>
> OSCAR (s=1)	0.0507	0.1102	0.3487	0.7623
>
> I think for this rate point, **you could not claim OSCAR achieves performance competitive with DiffEIC (DiffEiC use less bits but achieve much better DISTS) and significantly surpasses PerCo (You should compare the low-resolution datasets) in both perceptual and pixel-level fidelity.**
>
> **About Q6: Abnormal RD-Curves, which I think is the biggest concern.**
>
> **I think the simplest analysis is to test more rate points to show the RD-Curve with more details**(Since your model supports variable bitrates). Non-convex or Non-concave curves seems to show that your solution may show performance drops (e.g., PSNR degradation) at certain bitrates, which this indicates a fundamental instability in the model’s rate control.
> **It is necessary for variable bitrate models**: As a variable bitrate model, it must ensure smooth performance transitions when the bitrate changes continuously. If performance in certain intervals is significantly worse than that of adjacent bitrate points (even at the same downsampling ratio), this should be attributed to model design (such as the codebook allocation strategy or downsampling-reconstruction coupling).
>
> Finally, about Q4:
> I think the technique of existing variable-bitrate coding, which usually modulate the latent with a scaler does not conflict with your mechanism that use different diffusion steps. It is just a suggestion that you may could have a try in the future, but not a necessity.

---

> > ### Author Response · Authors · 2025-08-03
> >
> > Thank you for your thoughtful feedback and suggestions regarding both the multi-step diffusion comparison and the RD-curve analysis. We have carefully reviewed these points and provide our clarifications below.
> >
> > - Q1: Further discussion with multi-step diffusion codecs
> >
> >     We notice that you claim OSCAR underperforms DiffEIC at the 0.05 rate point. However, OSCAR actually achieves better results across all three metrics at this point. In your table, the bpp values appear to be aligned with the DISTS column, which might have led to a misunderstanding. We kindly request you to double-check this point. We will also add visual comparisons in the revised paper to further support this point.
> >
> >     Regarding PerCo, while their paper presents experiments on medium-sized images (e.g., 768$\times$512), they do not state that the method is inherently limited to low-resolution settings. Nonetheless, to address your concern, we provide results on the Kodak dataset, which has images with low resolution.
> >
> >     | models | bpp | DISTS↓ | LPIPS↓ | MS-SSIM↑ |
> >     | --- | --- | --- | --- | --- |
> >     | PerCo (s=20) | 0.0507 | 0.1410 | 0.3569 | 0.7012 |
> >     | DiffEIC (s=50) | 0.0422 | 0.1313 | 0.3427 | 0.7519 |
> >     | OSCAR (s=1) | 0.0507 | 0.1156 | 0.3361 | 0.7610 |
> >     | PerCo (s=20) | 0.0937 | 0.0993 | 0.2769 | 0.8081 |
> >     | DiffEIC (s=50) | 0.0945 | 0.0786 | 0.2352 | 0.8592 |
> >     | OSCAR (s=1) | 0.0937 | 0.0766 | 0.2340 | 0.8585 |
> >
> >     As shown, OSCAR also delivers competitive performance compared with DiffEIC and consistently outperforms PerCo on this low-resolution dataset.
> >
> > - Q2: Further discussion on RD-curves.
> >
> >     Thank you for the helpful suggestion. We would like to iterate that although the RD curve of OSCAR is not strictly convex, its overall trend remains stable and does not significantly deviate from a convex pattern. To support this point, we finetuned OSCAR on additional bitrates and provided results on the Kodak dataset for a finer-grained RD curve analysis. The results show no significant performance drop at any intervals.
> >
> >     | models | bpp | PSNR↑ | MS-SSIM↑ | DISTS↓ | LPIPS↓ |
> >     | --- | --- | --- | --- | --- | --- |
> >     | OSCAR | 0.0313 | 20.3616 | 0.7213 | 0.1287 | 0.3772 |
> >     | OSCAR | 0.0430 | 20.8756 | 0.7448 | 0.1208 | 0.3529 |
> >     | OSCAR | 0.0507 | 21.2287 | 0.7594 | 0.1154 | 0.3365 |
> >     | OSCAR | 0.0625 | 21.8956 | 0.7862 | 0.1047 | 0.3042 |
> >     | OSCAR | 0.0781 | 22.6267 | 0.8246 | 0.0904 | 0.2656 |
> >     | OSCAR | 0.0937 | 23.2489 | 0.8579 | 0.0788 | 0.2359 |
> >     | OSCAR | 0.1250 | 23.9707 | 0.8759 | 0.0693 | 0.2142 |
> >
> >     Since additional figures are not allowed this year, we also provide a simple Python code to visualize the trend of all these metrics, including PSNR, MS-SSIM, DISTS, and LPIPS.
> >
> >     ```python
> >     import matplotlib.pyplot as plt
> >
> >     bpp = [0.0313, 0.0430, 0.0507, 0.0625, 0.0781, 0.0937, 0.1250]
> >     metrics = {
> >         'PSNR':   [20.3616, 20.8756, 21.2287, 21.8956, 22.6267, 23.2489, 23.9707],
> >         'MS-SSIM':[0.7213, 0.7448, 0.7594, 0.7862, 0.8246, 0.8579, 0.8759],
> >         'DISTS':  [0.1287, 0.1208, 0.1154, 0.1047, 0.0904, 0.0788, 0.0693],
> >         'LPIPS':  [0.3772, 0.3529, 0.3365, 0.3042, 0.2656, 0.2359, 0.2142],
> >     }
> >
> >     fig, axes = plt.subplots(2, 2, figsize=(8, 6))
> >     for ax, (name, vals) in zip(axes.flat, metrics.items()):
> >         ax.plot(bpp, vals, 'o-')
> >         ax.set_xlabel('bpp')
> >         ax.set_ylabel(name)
> >         ax.set_title(f'{name} - bpp')
> >
> >     plt.tight_layout()
> >     plt.show()
> >     ```
> >
> >
> > Thank you again for your advice and the opportunity to engage in this valuable discussion. We are happy to explore how existing variable-bitrate techniques, such as latent scaling, can be integrated into our OSCAR in our immediate future work. We hope these results can address your remaining concerns.

---

> > > ### Comment · Reviewer_nsjq · 2025-08-04
> > >
> > > Thanks for your feedback. Firstly, I am sorry for my misunderstanding on the table. Your additional results on the Kodak dataset show that the proposed method remains stable across different bitrates, which address my earlier concerns. The overall quality of this paper is good and I will give a higher score.

---

> > > > ### Author Response · Authors · 2025-08-04
> > > >
> > > > Thank you very much for acknowledging our work and raising the score, as well as for the insightful suggestions and valuable discussions. We will continue to refine our work and add additional results based on your valuable feedback.

---

### Official Review · Reviewer_2faj · 2025-06-29

**Clarity:** 3
**Significance:** 2
**Originality:** 2
**Rating:** 4
**Confidence:** 4

**Summary:**

This paper proposes to use latent diffusion models for image compression. First, a set of hyper-encoders are trained to quantize the latent representation of any given image, where each hyper-encoder corresponds to a pre-defined compression bit-rate. The resulting quantized representation of an image is treated as its noisy version, in the form of the forward diffusion equation. The diffusion model is then fine-tuned with LoRA to take any such representation and "denoise" it in one-step, such that the resulting output minimizes a loss comprised of several similarity measure to the ground-truth non-quantized image representation. The input time-step to the diffusion model is learned from the bit-rate. Namely, a learned mapping takes the given bit-rate and outputs the corresponding ideal time-step, essentially translating lower bit-rates to larger time-steps (the representation is more "noisy" for lower bit-rates). The resulting compression algorithm demonstrates good rate-perception-distortion tradeoffs, outperforming previous methods while doing so only with one-step denoising at inference time.

**Questions:**

I have no questions for the authors.

**Ethical Concerns:**

["NO or VERY MINOR ethics concerns only"]

**Final Justification:**

The authors addressed most of my concerns and I think this paper is worthy of publication. The authors should add additional comparisons with recent methods, as discussed. This is especially important since the proposed method is not theoretically grounded and based only on evidence. It may be interesting to investigate in the future the reasons behind the effectiveness of the approach, but for the time being, this is a borderline paper for me.

**Limitations:**

No limitations are discussed in the main text.

Here are some of the limitations I noticed:
1. The mathematical motivation for the proposed approach is not well-explained in the paper. Yet the results are convincing.
2. The method relies heavily on a large pre-trained diffusion model, thus requiring a large memory footprint.
3. The ablation study is limited. I would personally be curious to see a toy experiment comparing the proposed approach to end-to-end neural compression, seeing whether the use of a pre-trained denoiser is a key ingredient in the success of the method.

**Paper Formatting Concerns:**

I do not have any formatting concerns.

**Quality:**

3

**Strengths And Weaknesses:**

The paper is overall well written, highly detailed, and provides comprehensive and convincing experiments and results. Compared to previous diffusion-based compression methods, the proposed algorithm only requires one-step denoising during inference, making the proposed approach much faster (as depicted in Fig. 2). However, the proposed algorithm still relies on a heavy latent-based diffusion model that contains many parameters, making the algorithm still not that practical for edge devices. Though, this is not to say that the gained speed-up compared to previous diffusion models is not an important step to take.

The biggest weakness in my opinion is the loose mathematical explanation that motivates the proposed method. The authors say that they "bridge" quantization and forward diffusion, yet this is explained only briefly in the paper and in very loose terms and claims (see e.g. L49). While the results are convincing and promising, it would be better if the authors discuss this idea in the main text in more detail.

Another weakness is that it's unclear what is the true role of diffusion here. There is no diffusion involved in the proposed method, but rather only the large pre-trained denoiser - which is fine-tuned. This raises the question whether the "diffusion" idea behind the proposed method is even significant in the first place - and I believe that it is not. The proposed approach can be considered an end-to-end neural compression method that uses vector quantization in the middle, with particular inductive biases enforced along the way. This raises the important question whether the method works well simply due to such inductive biases, but this is not explored in the paper.

Lastly, the main compared competing method is PerCo. However, to the best of my knowledge, there are other diffusion-based compression methods that outperform PerCo, such as [1] and [2]. I suggest that the authors add more comparisons with recent methods to their manuscript.

[1] Xu et al. "Idempotence and Perceptual Image Compression." In ICLR 2024.

[2] Ohayon et al. "Compressed Image Generation with Denoising Diffusion Codebook Models." In ICML 2025.

---

> ### Author Rebuttal · Authors · 2025-07-30
>
> Thanks for your valuable comment, nice suggestions, and for acknowledging our writing and empirical performance. Your questions and suggestions are instrumental in further strengthening our paper. Below, we respond to your specific comments.
>
> - Q1: The mathematical explanation and the motivation could be discussed in more detail.
>
>     Thank you for this insightful comment. We acknowledge that the core of OSCAR is based on an empirical assumption. While this assumption is not derived from a rigorous mathematical formulation, it is strongly supported by experimental evidence.
>
>    **The motivation behind OSCAR is as follows:** We begin by recalling the standard formulation of the forward diffusion process: $z_t=\sqrt{\alpha_t} z_0+\sqrt{1-\alpha_t}\epsilon$, where  $\epsilon \sim \mathcal{N}(0,1)$. Since we aim to use the diffusion formulation to reconstruct the clean latent $z_0$ from the quantized latent $\tilde{z}$, we naturally ask the question, can $\tilde{z}$ also be expressed in the form $\tilde{z}=\sqrt{\alpha} z_0+\sqrt{1-\alpha}\epsilon$? We test different values of $\alpha$ and visualize distribution of the quantization error, computed as $\epsilon=(\tilde{z}-\sqrt{\alpha} z_0)/ (\sqrt{1-\alpha})$. Interestingly, we find that after the first-stage alignment, it is possible to select a suitable $\alpha$ such that the resulting quantization error closely approximates a Gaussian distribution, as illustrated in Figure 4. For a fixed bitrate, we find that the selected $\alpha$ values are similar across different images. This motivates us to statistically determine the most appropriate $\alpha$ for each bitrate. Specifically, we compute the average cosine similarity across samples in our training set and select the pseudo timestep $t$ whose theoretical cosine similarity $\sqrt{\bar{\alpha}_t}$ best matches the observed value. This mapping procedure ensures a stable and interpretable assignment of bitrates to diffusion timesteps.
>
>     In the introduction section L45–59, we outline the high-level motivation. In Section 3.4, we provide detailed formulations of how we map the bitrates to pseudo diffusion timesteps. Appendix C provides a mathematical justification for the validity of using cosine similarity to establish the mapping. We will consolidate these discussions in the main text to provide a more coherent explanation.
>
> - Q2: Whether the method's effectiveness stems from inductive biases rather than the use of diffusion.
>
>     We would like to clarify the role of diffusion in our method. In Section 3.1, we provide the mathematical formulation of one-step diffusion. The only adaptation we introduce in OSCAR is treating the quantized latent as the noisy input to the denoising process—no changes are made to the core diffusion architecture or its operations. This design remains fully consistent with the one-step diffusion paradigm.
>
>     To demonstrate the role of diffusion, we removed the time module in the diffusion backbone and let the diffusion backbone predict the clean latent directly (thus making the backbone a pure denoiser). We trained this denoiser with the same experimental setting as OSCAR. Below, we report the performance on the DIV2K-val dataset.
>
>     | models | bpp | DISTS↓ | LPIPS↓ | MS-SSIM↑ |
>     | --- | --- | --- | --- | --- |
>     | OSCAR | 0.0313 | 0.1262 | 0.3883 | 0.7210 |
>     | Denoiser | 0.0313 | 0.1687 | 0.4259 | 0.6761 |
>     | OSCAR | 0.0507 | 0.1102 | 0.3487 | 0.7623 |
>     | Denoiser | 0.0507 | 0.1479 | 0.3726 | 0.7254 |
>     | OSCAR | 0.0937 | 0.0694 | 0.2385 | 0.8713 |
>     | Denoiser | 0.0937 | 0.0933 | 0.2697 | 0.8224 |
>     | OSCAR | 0.1250 | 0.0617 | 0.2157 | 0.8906 |
>     | Denoiser | 0.1250 | 0.0887 | 0.2514 | 0.8504 |
>
>     The diffusion formulation shows a clear advantage over the denoiser. We attribute this to our design strictly following the one-step diffusion framework, thus allowing us to fully leverage pre-trained diffusion backbones.
>
> - Q3: Comparison with more recent baselines.
>
>     Thank you for the advice. We will supplement these comparisons in our paper, as well as a discussion of similar works in the related work section.
>
>     1. Comparison with IPIC.
>
>         The table below presents results on the Kodak dataset.
>
>         | models | bpp | DISTS↓ | LPIPS↓ | MS-SSIM↑ |
>         | --- | --- | --- | --- | --- |
>         | IPIC (s=1000) | 0.0721 | 0.1674 | 0.3885 | 0.8221 |
>         | OSCAR (s=1) | 0.0507 | 0.1156 | 0.3360 | 0.7610 |
>         | IPIC (s=1000) | 0.1200 | 0.1640 | 0.2811 | 0.8447 |
>         | OSCAR (s=1) | 0.1250 | 0.0766 | 0.2340 | 0.8585 |
>
>         The results show that OSCAR performs better than IPIC on DISTS and LPIPS, while IPIC shows better MS-SSIM at low bitrates. This could be attributed to the generative nature of the diffusion backbone, which prioritizes perceptual quality over pixel-wise reconstruction.
>
>     2. Comparison with DDCM.
>
>         In our method, we use LPIPS-AlexNet as a training loss, and LPIPS-VGG as the evaluation metric to ensure a fair comparison. Here we follow DDCM to use LPIPS-AlexNet (denoted as LPIPS*), PSNR, and FID for evaluation, and use the DIV2K-val dataset with images cropped to 768$\times$768.
>
>         | model | bpp | PSNR↑ | LPIPS*↓ | FID↓ |
>         | --- | --- | --- | --- | --- |
>         | DDCM (s=1000) | 0.0420 | 21.1670 | 0.2010 | 18.3300 |
>         | OSCAR (s=1) | 0.0313 | 21.3304 | 0.1958 | 31.8673 |
>         | DDCM (s=1000) | 0.0660 | 21.7800 | 0.1720 | 16.3960 |
>         | OSCAR (s=1) | 0.0507 | 21.9478 | 0.1554 | 28.9917 |
>         | DDCM (s=1000) | 0.1370 | 22.7070 | 0.1380 | 14.6620 |
>         | OSCAR (s=1) | 0.1250 | 23.4529 | 0.1097 | 15.7407 |
>
>         Compared to the 1000-step DDCM, our one-step OSCAR performs better on reference-based metrics (PSNR and LPIPS*), suggesting higher fidelity to the original images. While DDCM achieves lower FID scores, the gap narrows as the bitrate increases.
>
> - Q4: The method relies heavily on a large pre-trained diffusion model.
>
>     We appreciate your acknowledgment that, despite the reliance on a pre-trained diffusion model, our one-step diffusion design takes an important step to reduce the time burden. Here we would like to supplement that leveraging pre-trained diffusion models has become a common practice in recent works—PerCo, DiffEIC, and DDCM all rely on them. Compared to these multi-step diffusion codecs, our one-step approach not only greatly reduces computational overhead but also achieves competitive or even better performance.
>
>
> ---
>
> We hope this response can help address your concerns. We believe our work can have an important impact on the field of image compression by offering a new perspective on leveraging pre-trained diffusion models for efficient one-step reconstruction across multiple bitrates. If you find our response satisfactory, we would be sincerely grateful if you could consider a higher score, as it would greatly support our efforts. Thank you once again for your consideration.

---

> > ### Comment · Reviewer_2faj · 2025-08-02
> >
> > I thank the authors for their response. I still stand by my original response: The method works well and the paper is well written, but it lacks sufficient theoretical motivation. It is based purely on evidence, as the authors say as well.
> >
> > Regarding the new comparisons with DDCM and IPIC: These seem compelling. I think it is important to emphasize in the paper the differences between the methods, when each method is better than the other, etc.
> >
> > I therefore keep my original score.

---

> > > ### Author Response · Authors · 2025-08-02
> > >
> > > Thank you for acknowledging the effectiveness of our method and our writing. We will continue refining our work to make a meaningful contribution to the community. However, we would like to emphasize the soundness and validity of our perspective on mapping bitrates to pseudo diffusion timesteps, which we have validated at two levels: (i) the assumption level — showing that the quantization error closely follows a Gaussian distribution at an appropriate pseudo timestep; and (ii) the system level — demonstrating that OSCAR achieves strong one-step, multi-bitrate performance against competitive baselines. We believe this work presents a valuable insight and will inspire future research.
> > >
> > > As for the new baselines, DDCM formulates the denoising process as selecting codebook indices. IPIC is a perceptual image codec that inverts an unconditional generative model with idempotence constraints. Both methods require up to 1000 reconstruction steps. As discussed in the rebuttal, OSCAR achieves competitive performance and, in most cases, superior results using only a single step. We will integrate these discussions and results into the revised paper. Thank you again for your support and consideration.

---

### Official Review · Reviewer_cDUu · 2025-06-30

**Clarity:** 4
**Significance:** 3
**Originality:** 3
**Rating:** 4
**Confidence:** 4

**Summary:**

This paper proposes OSCAR, a method for restoring images from latent compressed at different bitrates by constructing a mapping from compression bitrate to pseudo diffusion denoising timesteps, enabling a single-step reconstruction using one unified diffusion model.

**Questions:**

1.	As mentioned in the weaknesses regarding generalization to continuous bitrates, could the authors provide constructive insights or perspectives on how this method could be extended to support continuous bitrate adaptation in future work?
2.	The authors mention that a dedicated hyper-encoder is trained for each bitrate. Although the parameter count for each is relatively small, this design choice appears somewhat ad hoc. Would it be possible to allow different hyper-encoders to share modules? If so, would such sharing significantly affect the effectiveness of the proposed method?

**Ethical Concerns:**

["NO or VERY MINOR ethics concerns only"]

**Final Justification:**

My concerns and questions were answered.

**Quality:**

3

**Strengths And Weaknesses:**

Strengths:

1.	This paper adopts a novel and insightful perspective by constructing a mapping network from bitrate to pseudo-timestep, proposing an efficient one-step diffusion reconstruction method. This approach significantly accelerates the decoding process and enhances cross-bitrate generalization, thereby greatly reducing training and deployment costs. The methodology is supported by a rigorous and well-structured theoretical analysis.
2.	The proposed staged training pipeline and the design of the loss functions are clearly motivated and well-aligned with the task objectives. Extensive experiments demonstrate that OSCAR outperforms existing methods across multiple datasets, including Kodak, DIV2K-val, and CLIC2020, achieving superior performance in terms of computational efficiency, structural fidelity, and perceptual quality.

Weakness:

1.	The core idea of this work relies on a key assumption: that at high bitrates, the quantization error of the latent representations can be approximated as additive Gaussian noise. However, this assumption does not hold at low bitrates. Although the paper introduces optimized training for representation alignment across various bitrates, these bitrates are still pre-defined and discrete. The generalization ability of the model to unseen or non-predefined bitrates is not discussed.
2.	The design perspective adopted by the authors is highly insightful. However, although a fitting is performed between the cosine similarity and the diffusion timestep, the resulting mapping function is essentially based on empirical regression, lacking explicit theoretical constraints and cross-model generalization analysis. This absence of theoretical grounding may ultimately limit the method’s ability to generalize across continuously varying compression bitrates.
3.	In this work, a separate hyper-encoder is trained for each quantization bitrate, which limits the method’s ability to generalize to continuous or finer-grained bitrate settings.

---

> ### Author Rebuttal · Authors · 2025-07-30
>
> Thank you for your valuable comments and insightful suggestions, as well as for acknowledging our novelty, motivation, and convincing evaluations. Below, we respond to your comments.
>
> - Q1: The authors should discuss the generalization ability to unseen or non-predefined bitrates.
>
>     Thanks for the suggestion. Since our method employs a learnable hyper-encoder for compression, it does not support unseen bitrates out of the box. However, we conducted additional experiments to demonstrate OSCAR’s ability to quickly adapt to new bitrates with minimal computational overhead. We adopted a simple training strategy: half of the steps are allocated to the new bitrate, and half to the original groups. Extending to a new bitrate takes only 15K steps—under 4 hours on 4×A6000 GPUs. We attribute this efficiency to the fact that the diffusion model has already encountered structurally similar latent distributions during initial training, enabling fast and stable adaptation to nearby bitrates.
>
>     The tables below report results on the DIV2K-val dataset, where we progressively adapt the model to new bitrates one at a time. Numbers in parentheses indicate the performance change compared to the model before adaptation to the new bitrate.
>
>     | bpp | DISTS↓ | LPIPS↓ | MS-SSIM↑ |
>     | --- | --- | --- | --- |
>     | 0.0313 | 0.1262 | 0.3883 | 0.7210 |
>     | 0.0507 | 0.1102 | 0.3487 | 0.7623 |
>     | 0.0937 | 0.0694 | 0.2385 | 0.8713 |
>     | 0.1250 | 0.0617 | 0.2157 | 0.8906 |
>
>     | bpp | DISTS↓ | LPIPS↓ | MS-SSIM↑ |
>     | --- | --- | --- | --- |
>     | 0.0313 | 0.1306 (↑3.49%) | 0.3927 (↑1.13%) | 0.7161 (↓0.68%) |
>     | 0.0507 | 0.1141 (↑3.54%) | 0.3523 (↑1.04%) | 0.7578 (↓0.59%) |
>     | 0.0937 | 0.0728 (↑4.90%) | 0.2427 (↑1.77%) | 0.8595 (↓1.35%) |
>     | 0.1250 | 0.0648 (↑4.93%) | 0.2198 (↑1.91%) | 0.8766 (↓1.57%) |
>     | **0.0430 (new)** | 0.1198 | 0.3670 | 0.7402 |
>
>     | bpp | DISTS↓ | LPIPS↓ | MS-SSIM↑ |
>     | --- | --- | --- | --- |
>     | 0.0313 | 0.1360 (↑4.19%) | 0.4006 (↑2.03%) | 0.7031 (↓1.81%) |
>     | 0.0507 | 0.1185 (↑3.87%) | 0.3589 (↑1.87%) | 0.7459 (↓1.57%) |
>     | 0.0937 | 0.0752 (↑3.34%) | 0.2457 (↑1.25%) | 0.8506 (↓1.03%) |
>     | 0.1250 | 0.0672 (↑3.75%) | 0.2228 (↑1.38%) | 0.8664 (↓1.16%) |
>     | 0.0430 | 0.1246 (↑4.01%) | 0.3742 (↑1.98%) | 0.7282 (↓1.62%) |
>     | **0.0781 (new)** | 0.0918 | 0.2766 | 0.8412 |
> - Q2: The assumption of the method might not hold at low bitrates.
>
>     We would like to clarify that although the cosine similarity between quantized and clean latents naturally decreases at lower bitrates, the assumption that the quantization error follows additive Gaussian noise still holds, as long as the cosine similarity remains stable during the second-stage training.
>
>     To further address your concern, we retrained OSCAR from scratch to include two additional lower bitrates using the same experimental settings, except that we increased the training iterations to 150k. The results show that our method remains effective at these lower rates. We also visualize the quantization error using the same method as in Figure 4, and the distribution continues to closely follow a Gaussian. These results will be included in the revised manuscript.
>
>     The tables below report results on the DIV2K-val dataset. Numbers in parentheses indicate the performance change relative to our original model.
>
>     | models | bpp | DISTS↓ | LPIPS↓ | MS-SSIM↑ |
>     | --- | --- | --- | --- | --- |
>     | OSCAR | 0.0313 | 0.1274 (↑0.95%) | 0.3866 (↓0.44%) | 0.7234 (0.33%↑) |
>     | OSCAR | 0.0507 | 0.1098 (↓0.36%) | 0.3479 (↓0.23%) | 0.7644 (0.28%↑) |
>     | OSCAR | 0.0937 | 0.0702 (↑1.15%) | 0.2398 (↑0.55%) | 0.8694 (↓0.22%) |
>     | OSCAR | 0.1250 | 0.0630 (↑0.96%) | 0.2170 (↑0.60%) | 0.8881 (↓0.28%) |
>     | OSCAR | **0.0234 (new)** | 0.1494 | 0.4265 | 0.6993 |
>     | PerCo (SD) | 0.0234 | 0.2359 | 0.4433 | 0.6472 |
>     | OSCAR | **0.0098 (new)** | 0.1820 | 0.4628 | 0.6556 |
>     | PerCo (SD) | 0.0098 | 0.2870 | 0.4841 | 0.5889 |
> - Q3: Authors should provide more theoretical and cross-model generalization analysis.
>
>     Thanks for the valuable advice. We agree that OSCAR is based on an empirical assumption, which we validate through extensive experiments, rather than deriving from a strictly theoretical framework. While establishing a formal theoretical mapping is certainly an interesting direction, it remains challenging and still has a long way to go. In this context, we propose an alignment-based approach, where we empirically model the quantization error as Gaussian noise, based on the observed cosine similarity between the quantized and original latents. This assumption is strongly supported by our experiments. As shown in Figure 4, the quantization error closely follows a Gaussian distribution. We will include additional visualizations of quantization error at different bitrates in our paper. We believe that our perspective, supported by empirical findings, will provide valuable insights for future research.
>
>     To assess cross-model generalization, we replaced the UNet-based SD-2.1 with the Transformer-based SD-3 Medium and conducted experiments accordingly. We report the results on the DIV2K-val dataset. The results show that our method remains effective under this different backbone, demonstrating cross-model generalization.
>
>     | models | bpp | DISTS↓ | LPIPS↓ | MS-SSIM↑ |
>     | --- | --- | --- | --- | --- |
>     | OSCAR (UNet) | 0.0313 | 0.1262 | 0.3883 | 0.7210 |
>     | OSCAR (DiT) | 0.0313 | 0.1076 | 0.3579 | 0.7532 |
>     | OSCAR (UNet) | 0.0937 | 0.0694 | 0.2385 | 0.8713 |
>     | OSCAR (DiT) | 0.0937 | 0.0657 | 0.2124 | 0.9003 |
> - Q4: One hyper-encoder is trained for each bitrate, which might limit the generalization to fine-grained bitrate settings.
>
>     We agree that generalizing to fine-grained bitrate settings remains a challenge, especially when aiming for both flexibility and high performance. However, as shown in Appendix Figure 1, despite using separate hyper-encoders, the quantized latents of nearby bitrates remain highly similar, mirroring the behavior of variable-bitrate methods. We believe this latent consistency suggests that OSCAR has the potential for future improvements.
>
> - Q5: Would it be possible to allow different hyper-encoders to share modules?
>
>     We would like to clarify that each hyper-encoder is tied to a specific bitrate because each uses different internal settings, like the downsampling ratio and codebook size, to control the compression level. The module requires no complex design and is very lightweight ($\approx$0.5% of the parameters compared to the diffusion backbone). We found that having a separate lightweight hyper-encoder per bitrate works well and keeps things simple. Nevertheless, an interesting future direction is to share a single hyper-encoder across bitrates that use the same downsampling ratio but differ in codebook size. This would allow multiple bitrates to be paired with one hyper-encoder.
>
> - Q6: How can this method be extended to continuous bitrate adaptation in future work?
>
>     Thank you very much for the inspiring question. One important current limitation lies in the total training budget. Under constrained resources (e.g., 4×A6000 GPUs), it is challenging for a single model to fully optimize across many bitrates without sacrificing performance. A straightforward solution is to scale up training, e.g., using larger batch sizes and more iterations to allow the model to better generalize across bitrate levels.
>
>     Furthermore, our results (Appendix Figure 1) show that quantization errors tend to be similar when the downsampling ratios are the same. This suggests a promising future direction: we could train a model on several anchor bitrates with different downsampling ratios, and then generalize to additional bitrate levels, which share the same downsampling ratio as the anchor bitrate but with different codebook sizes. In fact, as discussed in Q1, our simple training strategy already allows fast adaptation to new bitrates with minimal cost, leveraging well-trained anchor bitrates. We believe that designing efficient and scalable mechanisms for bitrate extension is a very promising direction.
>
>
> ---
>
> We hope this response can help address your concerns. We believe our work can have an important impact on the field of image compression by offering a new perspective on leveraging pre-trained diffusion models for efficient one-step reconstruction across multiple bitrates. If you find our response satisfactory, we would be sincerely grateful if you could consider a higher score, as it would greatly support our efforts. Thank you once again for your consideration.

---

> ### Author Response · Authors · 2025-08-03
>
> Thank you for your acknowledgement. We hope our response has satisfactorily addressed your concerns, and we will incorporate these important empirical results and discussions into the revised version. May we kindly ask if there are any remaining concerns? We would be more than happy to address them. Thank you once again for your time and thoughtful consideration.

---

> > ### Comment · Reviewer_cDUu · 2025-08-04
> >
> > Thanks, my concerns were answered.

---

> > > ### Author Response · Authors · 2025-08-04
> > >
> > > Thanks for your time and effort in reviewing our paper. We are grateful for your feedback throughout this process. We will continue to refine our work and add additional results based on your valuable feedback.

---

### Note · Authors · 2025-08-11

Dear reviewers and area chairs,

We sincerely appreciate the reviewers’ time, valuable feedback, and constructive suggestions. We are pleased that all reviews express a positive inclination toward accepting our submission. Overall, the reviewers describe our paper as **well-written** (2faj), of **good quality** (nsjq), and supported by a **well-structured analysis** (cDUu). Our methodology is recognized as **novel**, **elegant**, and **clearly motivated** (cDUu, 7kzL), delivering **superior** (cDUu), **convincing** (2faj), **stable** (nsjq) and **promising** (7kzL) performance, while notably **improving efficiency** (cDUu, 2faj, nsjq, 7kzL).

We have addressed each reviewer’s comments individually and summarize our main responses as follows:

1. Clarified the **generalization ability** of OSCAR to unseen and non-predefined bitrates.
2. Demonstrated that OSCAR also achieves superior performance at **lower bitrates**.
3. Provided additional analysis and **cross-model generalization** to further explain and validate our motivation.
4. Highlighted the critical role of the **diffusion backbone** by comparing with an end-to-end neural compression design.
5. Added a detailed discussion of the **hyper-encoder design**, clarifying its advantages and potential extension to finer-grained bitrates.
6. Expanded the discussion of related work, particularly DDPM and IPIC, both of which are multi-step diffusion codecs.
7. Added **complexity comparisons**, including MAC counts and encoding-decoding time, against leading baselines.
8. Demonstrated that OSCAR produces **smooth RD-curves**, ensuring stable performance when bitrates change continuously.
9. Reported additional **PSNR and FID** results, showing OSCAR remains competitive.
10. Discussed the **limitations** of diffusion-based codecs and showed that the pixel-wise fidelity bottleneck originates from the VAE, which favors perceptual quality.

Again, we extend our gratitude to all reviewers and area chairs!

Best regards,

Authors

---

### Decision · Program_Chairs · 2025-09-17

**Decision:**

Accept (poster)

**Comment:**

OSCAR proposes an efficient one-step diffusion codec that supports multiple bitrates using a single model, achieving strong empirical results across datasets, bitrate settings, and model backbones. It performs competitively while being significantly faster than prior diffusion-based codecs.

were initially positive but raised concerns around generalization to unseen bitrates, lack of theoretical grounding, and unusual RD-curves. The authors responded thoroughly. They added new results on low-bitrate performance, RD-curve analysis, baselines, and visual comparisons. Reviewers acknowledged the updates, and all maintained or raised their positive scores after discussion.

While the method lacks deep theoretical grounding and still depends on pre-defined bitrate anchors, it offers a empirically valuable contribution. With the promised revisions, the AC recommends acceptance. This is an interesting and impactful paper—congrats to the authors for their work and clear rebuttal!